# Retrotransposons in Werner syndrome-derived macrophages trigger type I interferon-dependent inflammation in an atherosclerosis model

Sudip Kumar Paul [1], Motohiko Oshima[2], Ashwini Patil [3], Masamitsu Sone[1,9], Hisaya Kato [4], Yoshiro Maezawa[4], Hiyori Kaneko[4], Masaki Fukuyo[5], Bahityar Rahmutulla[5], Yasuo Ouchi[1,6], Kyoko Tsujimura[1], Mahito Nakanishi[7], Atsushi Kaneda [5], Atsushi Iwama [2], Koutaro Yokote[4] ✉, Koji Eto [1,8] ✉ & Naoya Takayama [1] ✉

The underlying mechanisms of atherosclerosis, the second leading cause of death among Werner syndrome (WS) patients, are not fully understood. Here, we establish an in vitro co-culture system using macrophages (iMφs), vascular endothelial cells (iVECs), and vascular smooth muscle cells (iVSMCs) derived from induced pluripotent stem cells. In co-culture, WS-iMφs induces endothelial dysfunction in WS-iVECs and characteristics of the synthetic phenotype in WS-iVSMCs. Transcriptomics and open chromatin analysis reveal accelerated activation of type I interferon signaling and reduced chromatin accessibility of several transcriptional binding sites required for cellular homeostasis in WS-iMφs. Furthermore, the H3K9me3 levels show an inverse correlation with retrotransposable elements, and retrotransposable element-derived double-stranded RNA activates the DExH-box helicase 58 (*DHX58*)-dependent cytoplasmic RNA sensing pathway in WS-iMφs. Conversely, silencing type I interferon signaling in WS-iMφs rescues cell proliferation and suppresses cellular senescence and inflammation. These findings suggest that Mφ-specific inhibition of type I interferon signaling could be targeted to treat atherosclerosis in WS patients.

Werner syndrome (WS) is a rare human inherited disorder characterized by the appearance of premature aging induced by mutation of the *WRN* (RecQ-like helicase) gene[1]. Atherosclerosis is frequently observed in WS patients and is their second leading cause of death after cancer[2,3]. Because a mouse model of WS does not recapitulate atherosclerosis[4], in vitro differentiation of atherosclerosis-associated cells from WS patient-specific pluripotent stem cells is an alternative technique for studying the mechanism of atherosclerosis in WS. However, a recent study reports that *WRN*-deficient embryonic stem cells (ESCs) that are differentiated into endothelial cells do not show features of premature aging[5]. Therefore, the pathogenesis of atherosclerosis in WS remains unclear.

Vascular aging is a major risk factor for the onset of cardiovascular diseases, which are the leading cause of death worldwide[6]. Macrophages (Mφs) have long been known to contribute to inflammatory diseases including atherosclerosis. They regulate local inflammation by secreting a wide array of pro-inflammatory cytokines and chemokines upon encountering various stimuli[7]. In particular, type I

interferon (IFN) signaling under the regulation of IFN regulatory factor 3 (IRF3) and IRF7 transcription factors is a trigger of metabolic change, mitochondrial dysfunction, cellular senescence, and aging[8–11]. Previous reports also suggest that type I IFN signaling-related genes are upregulated in human atherosclerotic plaques[12,13] and atherosclerosis-associated cells[14,15]. Moreover, IFN-β increases Mφ-endothelial cell adhesion, and silencing endogenous type I IFN protects against lesional accumulation of macrophages[12]. Thus, type I IFNs play a key role in the chronic inflammation associated with atherosclerosis independent of pathogens. Furthermore, growing evidence suggests that other than viruses, foreign factors, or IFN protein itself, cell-intrinsic factors such as reactive oxygen species (ROS), endoplasmic reticulum (ER) stress, DNA damage, and retrotransposable element (RTE) priming of type I IFN-dependent immune responses[8,16–19].

RTEs, which include long-interspersed nuclear elements (LINEs), short-interspersed nuclear elements (SINEs), and endogenous retroviruses (ERVs), comprise up to half of the human genome and usually remain silent by maintaining repressive heterochromatin formation across the genome[20–22]. However, in some pathogenic conditions, RTEs are reactivated and have the potential to promote aberrant transcription (which may lead to inflammation), alternative splicing, insertional mutagenesis, DNA damage, and genome instability[23]. Recent studies show that elevated human endogenous retroviral elements can initiate innate immune responses and cause vascular changes associated with pulmonary arterial hypertension[24,25]. In addition, the ability to repress RTE activity diminishes during the aging process[18,26]. In various primate and mouse aging models, de-repression of human ERVs provokes cellular senescence and tissue aging[27]. Moreover, RTEs have recently been associated with cardiovascular diseases, cancer, aging, and other cellular maintenance processes[18,24–26,28,29].

The in vitro generation of key players of atherosclerosis, including immune cells (e.g., Mφs) and vascular cells (e.g., vascular endothelial cells (VECs) and vascular smooth muscle cells (VSMCs)), from disease-specific human induced pluripotent stem cells (iPSCs) is a promising tool for mimicking compromised vessel walls[30]. Recently, we established disease-specific iPSCs from WS patients and repaired the genetic mutation in one allele of the *WRN* gene locus using CRISPR-Cas9 genome editing techniques to establish gene-corrected (gc)WS-iPSCs[31].

In this work, we generate, characterize, validate, and use immune and vascular cells differentiated from iPSCs from healthy individuals and WS patients, as well as gene-corrected iPSCs from WS patients, and establish an in vitro model to better understand the atherosclerosis microenvironment at the molecular level. WS-iMφs induce endothelial dysfunction in WS-iVECs and synthetic phenotype switching in WS-iVSMCs. RNA-sequencing (seq) and assay for transposase-accessible chromatin using sequencing (ATAC-seq) reveal accelerated type I IFN signaling and reduced chromatin accessibility in WS-iMφs. H3K9me3 inversely correlates with RTEs, activating the *DHX58*-dependent RNA sensing pathway. Silencing type I IFN signaling rescues proliferation and suppresses senescence and inflammation, suggesting atherosclerosis treatment in WS.

## Results

### Differentiation and characterization of Mφs and vascular cells from human iPSCs

We modified the revised human pluripotent stem cell-derived sac method[32–35] to generate functional Mφs (Supplementary Fig. 1a). Reverse transcription-quantitative polymerase chain reaction (RT-qPCR) analysis confirmed the mRNA expression of different Mφ maturation markers of iPSC-derived Mφs (iMφs) on day 21 of differentiation, whereas no such expression was detected in iPSCs or C3H10T1/2 feeder cells (Supplementary Fig. 2a). Also, like peripheral blood (PB)-Mφs, iMφs on day 21 of differentiation expressed higher levels of Mφ maturation cell surface markers (e.g., CD14, CD16, CD11b, HLA-DR, and CD163) than hematopoietic progenitor cells (HPCs) on

day 14 (Supplementary Fig. 2b). Giemsa-stained CD14⁺CD11b⁺ (fluorescence-activated cell sorting (FACS)-sorted) iMφs showed circular or irregular shapes with rich cytoplasm and a large nucleus (Supplementary Fig. 2c). Phagocytosis assay[36] confirmed the significant phagocytosis activity of iMφs (Supplementary Fig. 2d). In addition, when we treated iMφs with lipopolysaccharide (LPS) and assessed their immune response, we found dose-dependent mRNA expression of pro-inflammatory cytokines (Supplementary Fig. 2e).

We next used the same method to generate vascular progenitor cells (VPCs) on irradiated C3H10T1/2 feeder cells. These VPCs were further directed to differentiate into VECs or VSMCs in the presence or absence of vascular endothelial growth factor (VEGF), respectively (Supplementary Fig. 1b,c). After 7 days of differentiation from VPCs, iPSC-derived VECs (iVECs) positive for VE-cadherin (CD144), VEGFR2, CD34, and CD31 were induced (Supplementary Fig. 2f). We observed VEC-like morphologies (Supplementary Fig. 2g) and confirmed that iVECs expressed common markers of VECs (i.e., *VWF*, *CDH5*, *PECAM1*, and *NOS3*) at similar levels as human aortic endothelial cells (Supplementary Fig. 2h). In addition, immunocytochemistry confirmed that iVECs exhibited VE-cadherin protein at cell-to-cell junctions (Supplementary Fig. 2i) and uptake of acetylated low-density lipoprotein (Supplementary Fig. 2j). iPSC-derived VSMCs (iVSMCs) had VSMC-like morphologies (Supplementary Fig. 2k) and showed protein (Supplementary Fig. 2l) and gene (Supplementary Fig. 2m) expression of VSMC maturation markers. Collectively, these results demonstrate the successful induction of functional human iMφs, iVECs, and iVSMCs.

### Apoptosis and cellular senescence-dependent impaired cell proliferation capacity of WS-iMφs

After the successful induction of iPS-derived cells, we assessed the functionality of *WRN* gene correction. RT-qPCR revealed that gcWS-iMφs showed partial rescue of *WRN* mRNA expression to 51.22% of that in healthy-iMφs and an 8.73-fold increase compared with WS-iMφs (Fig. 1a). As WRN protein plays a critical structural role in optimizing DNA repair[37], we assessed γH2AX, the phosphorylated Ser-139 residue of the histone variant H2AX, which is an early cellular response to induction of DNA double-strand breaks[38]. FACS analysis showed significantly reduced γH2AX formation in gcWS-iMφs compared with WS-iMφs, which was confirmed by immunofluorescence (Fig. 1b,c).

We next investigated differences in induction efficiency, cell proliferation, and oxidized low-density lipoprotein (oxLDL) uptake among healthy-, WS-, and gcWS-iMφs, iVECs, and iVSMCs. No notable differences were observed between HPC and VPC generation or between iVEC and iVSMC proliferation (Supplementary Fig. 3a–c), as previously observed in *WRN* knock-out ESC-derived endothelial cells[5]. Moreover, we observed no difference in oxLDL uptake or foam cell formation among healthy-, WS-, and gcWS-iMφs (Supplementary Fig. 3d, e). However, WS-iMφs showed significantly lower proliferation capacity than healthy-iMφs, and proliferation was partially rescued in gcWS-iMφs (Fig. 1d). When we assessed whether the low proliferation of WS-iMφs was due to apoptotic cell death or cellular senescence before or after oxLDL treatment, we found that the proportion of annexin V⁺ cells was significantly higher among WS-iMφs than among healthy-iMφs but was significantly reduced in gcWS-iMφs (Fig. 1e). In addition, mRNA expression of the vital cell cycle inhibitor *CDKN1A*, which is strongly correlated with apoptotic cell death, was also significantly higher in WS-iMφs than in healthy- and gcWS-iMφs (Fig. 1f), indicating that *WRN* gene correction reduced apoptosis.

We further performed senescence assay using fluorescently labeled senescence-associated beta-galactosidase (SA-β-gal) among healthy-, WS-, and gcWS-iMφs. The mean fluorescence intensity (MFI) of SA-β-gal and mRNA expression of the senescence-associated gene *CDKN2A* was significantly higher for WS-iMφs than for healthy- and gcWS-iMφs irrespective of oxLDL treatment (Fig. 1g, h). Furthermore, in line with previous findings that senescent cells secrete a wide array

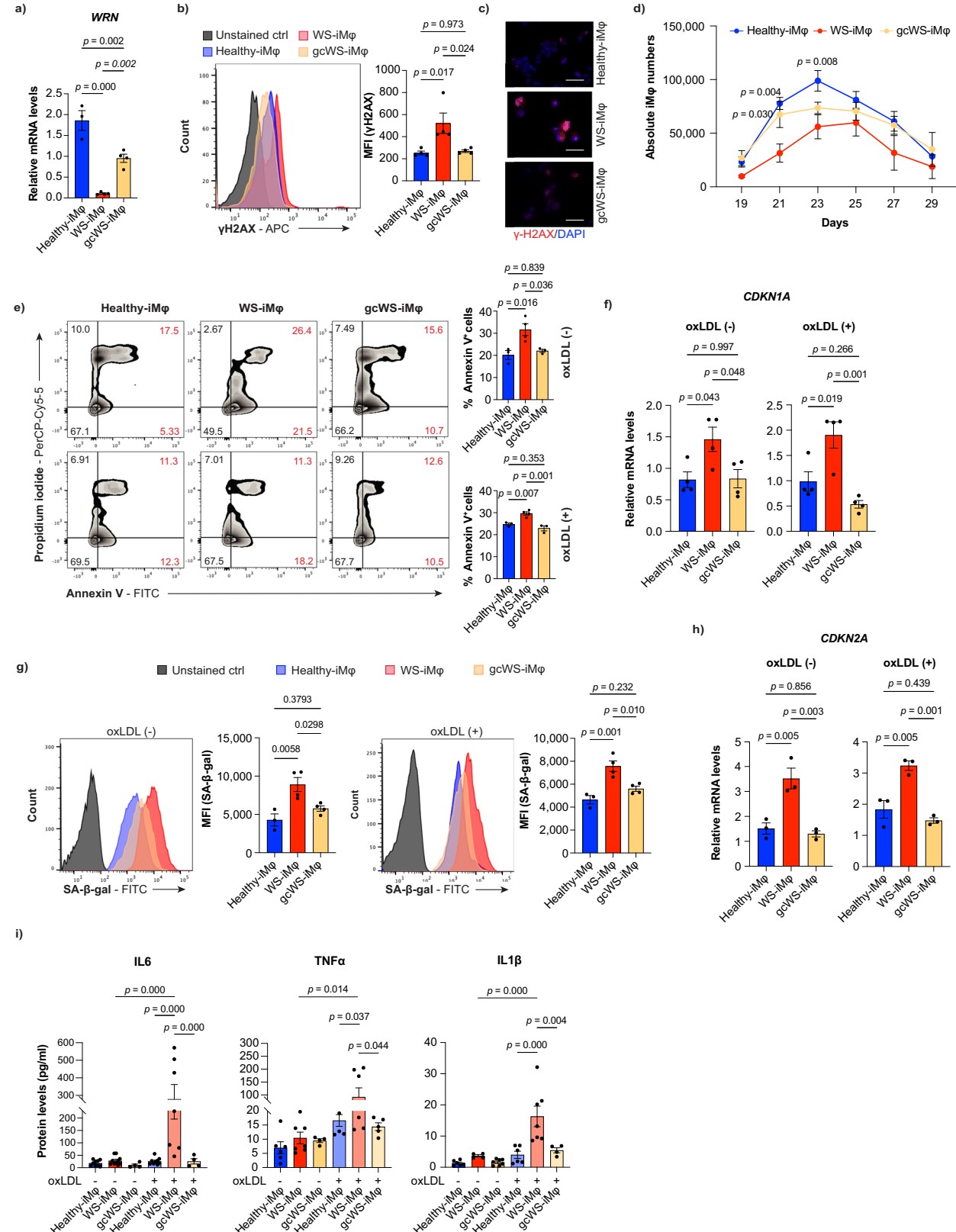

of pro-inflammatory cytokines, WS-iMφs showed more secretion of pro-inflammatory cytokines, including interleukin (IL)−6, IL-1β, and tumor necrosis factor-alpha (TNFα), especially after oxLDL treatment, whereas such secretion was significantly reduced by *WRN* gene correction (Fig. 1i).

## Recapitulation of early-stage atherosclerosis by co-culture of iMφs and iVECs or iVSMCs

Interactions among VECs, VSMCs, and the immune system play a vital role in the progression and outcome of cardiovascular diseases and atherosclerosis[39]. Therefore, to assess the effect of inflammatory iMφs

**Fig. 1 | Apoptosis and cellular senescence lead to impaired WS-iMφ proliferation. a** mRNA levels of *WRN* normalized by *GAPDH* mRNA (n = 4). **b** Representative flow cytometric histogram of γ-H2AX staining in healthy-, WS-, and gcWS-iMφs without oxLDL treatment (left) and γ-H2AX MFI (right) (n = 4). **c** Immunofluorescence image of γ-H2AX foci in healthy-, WS-, and gcWS-iMφs. The scale bar is 30 μm. **d** Absolute numbers of CD14⁺CD11b⁺ healthy- (n = 3), WS- (n = 4), and gcWS-iMφs (n = 3). **e** Representative flow cytometric plots of annexin V staining in healthy-, WS-, and gcWS-iMφs without (left top) or with (left bottom) oxLDL treatment. Bar graphs show the total proportion of annexin V⁺ cells among healthy- (n = 3), WS- (n = 4), and gcWS-iMφs (n = 3), before (right top) and after (right bottom) oxLDL treatment. **f** mRNA levels of *CDKN1A* normalized by *GAPDH* mRNA before (left) and after (right) oxLDL treatment (n = 4). **g** Representative flow cytometric plots of SA-β-gal staining before (left) and after (right) oxLDL treatment

among healthy-, WS-, and gcWS-iMφs. Bar graphs show the MFI of SA-β-gal among healthy- (n = 3), WS- (n = 4), and gcWS-iMφs (n = 4), before (left) and after (right) oxLDL treatment. **h** mRNA levels of *CDKN2A* normalized by *GAPDH* mRNA before (left) and after (right) oxLDL treatment (n = 3). **i** Secreted pro-inflammatory cytokine protein levels, determined by ELISA, for healthy-, WS-, and gcWS-iMφs before (n = 9, 10, 4) and after oxLDL treatment (n = 9, 7, 4). Three independent experiments of three independent biological samples were used for Healthy- ans WS-iMφs. Data are shown as the mean ± standard error of the mean (SEM) of biologically independent samples unless otherwise stated. One-way ANOVA with Tukey's multiple comparisons was performed to calculate the *p* values. MFI mean fluorescent intensity, oxLDL oxidized low-density lipoprotein. Source data are provided as a Source Data file.

on iVECs and iVSMCs in vitro, we performed a co-culture assay (Supplementary Fig. 4a).

Endothelial dysfunction is a hallmark of early-stage atherosclerosis and is characterized by increased Mφ adhesion, inflammation, and extracellular matrix degradation (Supplementary Fig. 4b). VEC-Mφ adhesion may be the initial stage of atherosclerotic plaque formation[40]. Accordingly, significantly more adherent Mφs were found in the WS group than in the healthy group (Fig. 2a). In line with this finding, we found significantly more ICAM-1⁺ iVECs and higher *ICAM1* mRNA levels in the WS group co-cultured with inflammatory WS-iMφs (Fig. 2b, c). In addition, we found significantly higher pro-inflammatory cytokine (IL6 and TNFα) and matrix metalloproteinase-1 (MMP1) protein secretion and mRNA expression in WS-iVECs than in healthy-iVECs when co-cultured with inflammatory WS-iMφs (Fig. 2d, e).

Next, we assessed the effect of iMφ co-culture on iVSMCs. The historical view of VSMCs in atherosclerosis is that the 'aberrant' proliferation of VSMCs promotes plaque formation[41]. The switching of VSMCs from a contractile phenotype to an adverse synthetic phenotype is a hallmark of atherosclerosis, which begins with high VSMC proliferation and reduced expression of VSMC contractile markers (Supplementary Fig. 4c)[42–44]. Specific markers that are upregulated in the synthetic phenotype are rare; instead, the disappearance of proteins associated with the contractile phenotype is generally considered characteristic of the synthetic phenotype[43]. Accordingly, we found that WS-iVSMCs were more highly proliferative than healthy-iVSMCs when co-cultured with inflammatory iMφs (Fig. 2f). Meanwhile, although we observed steady-state mRNA expression of VSMC markers in both healthy- and WS-iVSMCs, co-culture with inflammatory iMφs reduced mRNA levels of the VSMC contractile markers *CNN1*, *ACTA2*, *TAGLN*, and *SMTN* in WS-iVSMCs but not in healthy-iVSMCs (Fig. 2g). In addition, immunocytochemistry analysis revealed that calponin-1 protein and the MFI of calponin-1 expression decreased in WS-iVSMCs co-cultured with inflammatory iMφs, whereas this phenomenon was not observed in healthy-iVSMCs (Fig. 2h, i). Taken together, these results suggest that WS-iVSMCs are more prone to switching from a contractile phenotype to a more rigid synthetic phenotype.

Next, we sought to determine whether WS-iMφs or WS-derived vascular cells were responsible for generating the inflammatory atherosclerotic environment in co-culture. In a cross-co-culture experiment, we found that WS-iMφs were required to induce an inflammatory phenotype in iMφ-iVEC co-culture regardless of iVEC type (Supplementary Fig. 5a–d). On the other hand, only the combination of WS-iMφs and WS-iVSMCs was required to induce phenotypic changes in iVSMCs (Supplementary Fig. 5e, f). Next, we asked whether macrophage-vascular cell interactions or secretory elements from macrophages were required to induce an inflammatory phenotype of vascular cells. To answer this question, we used conditioned media (CM) from macrophage culture to cultivate mature vascular cells for 72 h. However, no notable changes were observed in the CM group compared with the control group (Supplementary Fig. 5g–k). These results suggest that macrophage-

vascular cell interactions are required to induce an inflammatory phenotype and that WS-iMφs orchestrate vascular health in WS.

## Upregulated type I IFN signaling pathway in WS-iMφs as revealed by RNA-seq analysis

To characterize the phenotype of WS-iMφs in more detail, we performed RNA-sequencing (RNA-seq) analysis in healthy-, WS-, and gcWS-iMφs with or without oxLDL treatment (Fig. 3a). In non-treated cells, compared with WS-iMφs, we identified 101 upregulated and 385 downregulated differentially expressed genes (DEGs) in healthy-iMφs and 433 upregulated and 866 downregulated DEGs in gcWS-iMφs. Whereas the differences between the numbers of upregulated and downregulated DEGs became smaller when we treated the cells with oxLDL (Fig. 3b). We next performed gene set enrichment analysis (GSEA) and plotted the top 10 significant pathways enriched along with the gene set in each pairwise analysis (Fig. 3c–i). GSEA revealed that several inflammatory pathways, including IFNα and γ response, and type I IFN signature gene sets were enriched in non-treated WS-iMφs compared with non-treated healthy- and gcWS-iMφs (Fig. 3c, e). Moreover, as expected from the finding that healthy- and gcWS-iMφs showed less apoptosis and senescence than WS-iMφs, GSEA showed that apoptosis and P53 pathways and gene sets were enriched in WS-iMφs compared with healthy- and gcWS-iMφs (Supplementary Data 1). By contrast, cell cycle and DNA repair-related pathways and gene sets were enriched only in healthy-iMφs compared with WS-iMφs (Fig. 3c). Furthermore, after oxLDL treatment, healthy-iMφs displayed some inflammatory and stress-responsive pathways along with the fatty acid metabolic pathway; however, WS-iMφs still exhibited higher expression of IFN signaling gene sets (Fig. 3d, Supplementary Data 1). The finding that no significant inflammatory pathways were enriched in gcWS-iMφs, regardless of oxLDL treatment, compared with WS-iMφs suggests that *WRN* gene correction suppressed IFN and inflammatory responses in gcWS-iMφs (Fig. 3e, f, Supplementary Data 1). Additionally, we noted that in response to oxLDL, healthy-, WS-, and gcWS-iMφs exhibited increases in IFN response genes and other genes associated with inflammation and decreases in genes related to the cell cycle (Fig. 3g–i, Supplementary Data 1). Interestingly, IFN signature gene expression did not change drastically in WS-iMφs, in contrast to healthy- and gcWS-iMφs (Fig. 3g–i).

Collectively, these results suggest that IFNα response-related and inflammatory gene sets were enriched in WS-iMφs regardless of oxLDL treatment and that *WRN* gene correction suppressed inflammatory pathways in gcWS-iMφs. When we performed RNA-seq analysis of healthy and WS-iPS-derived iVECs and iVSMCs, we found no upregulation of IFN signal genes or inflammation-related pathways regardless of oxLDL treatment (Supplementary Fig. 6a–j), further confirming that the culprit of atherosclerosis in WS is iMφs.

## Type I IFN-specific chromatin accessibility profile of WS-iMφs

To better understand WS-related epigenetic remodeling in iMφs, we performed open chromatin analysis via transposase-accessible

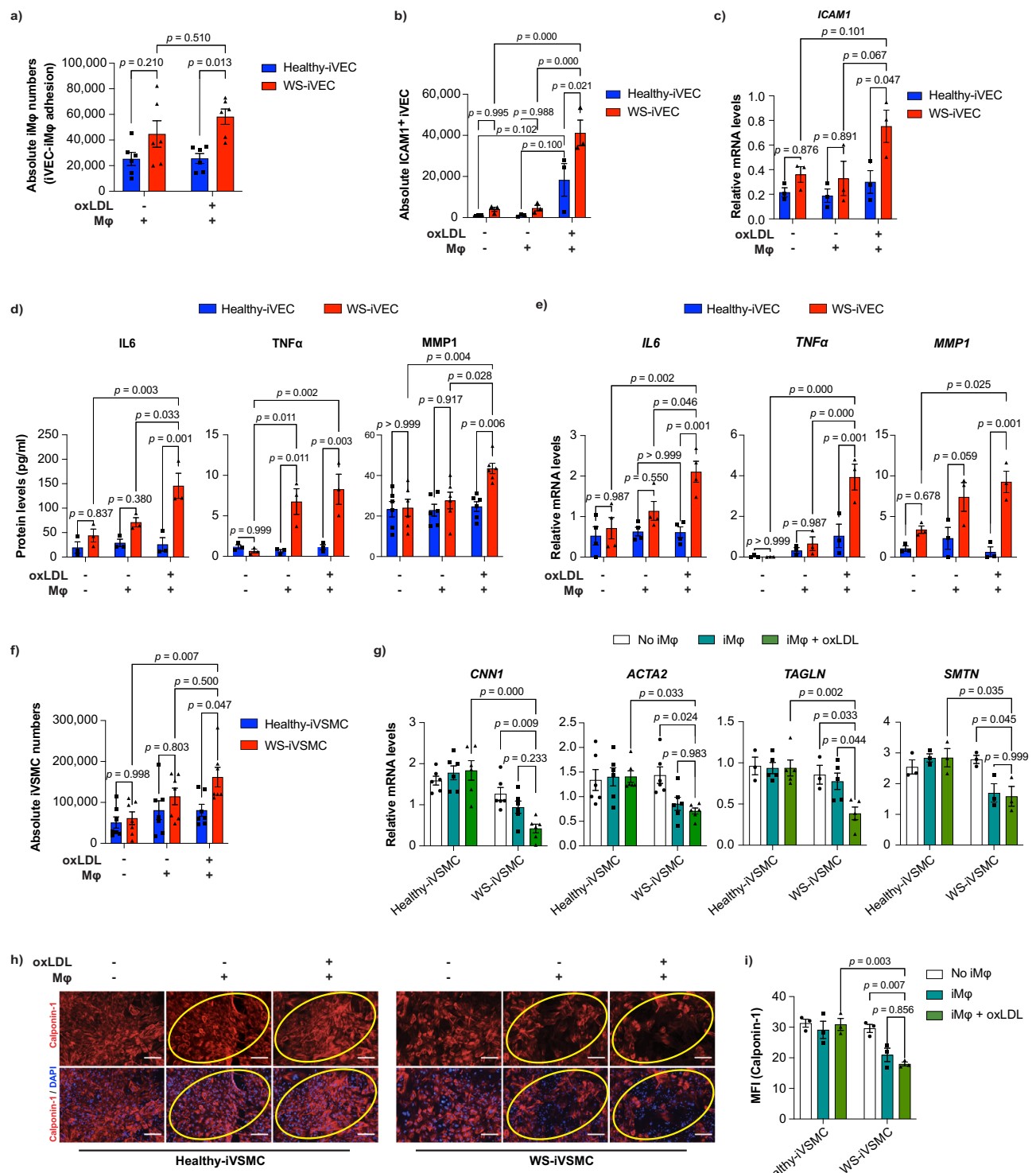

**Fig. 2 | Effects of inflammatory iMφs on iPSC-derived vascular cells. a** Absolute numbers of CD14[+] adherent iMφs on iVECs after co-culture (n = 6). **b** Absolute numbers of ICAM1[+] VE-cad[+] iVECs before and after co-culture with untreated or oxLDL-treated Mφs (n = 3). **c** mRNA levels of *ICAM1* normalized by *GAPDH* mRNA before and after co-culture with untreated or oxLDL-treated Mφs (n = 3). **d** IL6 (n = 3), TNFα (n = 3), and MMP1 (n = 6) protein levels quantified by ELISA before and after co-culture with untreated or oxLDL-treated Mφs. **e** *IL6*, *TNFα*, and *MMP1* mRNA levels normalized by *GAPDH* mRNA before and after co-culture with untreated or oxLDL-treated Mφs (n = 3). **f** Absolute numbers of healthy- and WS-iVSMCs before and after co-culture with untreated or oxLDL-treated Mφs (n = 7). **g** mRNA levels of VSMC contractile markers (*CNN1* (n = 6), *ACTA2* (n = 6), *TAGLN*

(n = 3, 5, 5), *SMTN* (n = 3)) normalized by *GAPDH* mRNA before and after co-culture with untreated or oxLDL-treated Mφs. **h** Immunocytochemistry of calponin-1 protein in healthy- and WS-iVSMCs before and after co-culture with untreated or oxLDL-treated Mφs. The scale bar is 100 μm. **i** MFI of calponin-1 protein expression from (**h**). ImageJ was used to calculate calponin-1 MFI (n = 3). Data are shown as the mean ± SEM. (n = 6) represents two biologically independent samples over two independent experiments, and (n = 3) represents biologically independent samples. Two-way ANOVA with Tukey's multiple comparisons was performed to calculate the *p* values. MFI mean fluorescent intensity, oxLDL oxidized low-density lipoprotein. Source data are provided as a Source Data file.

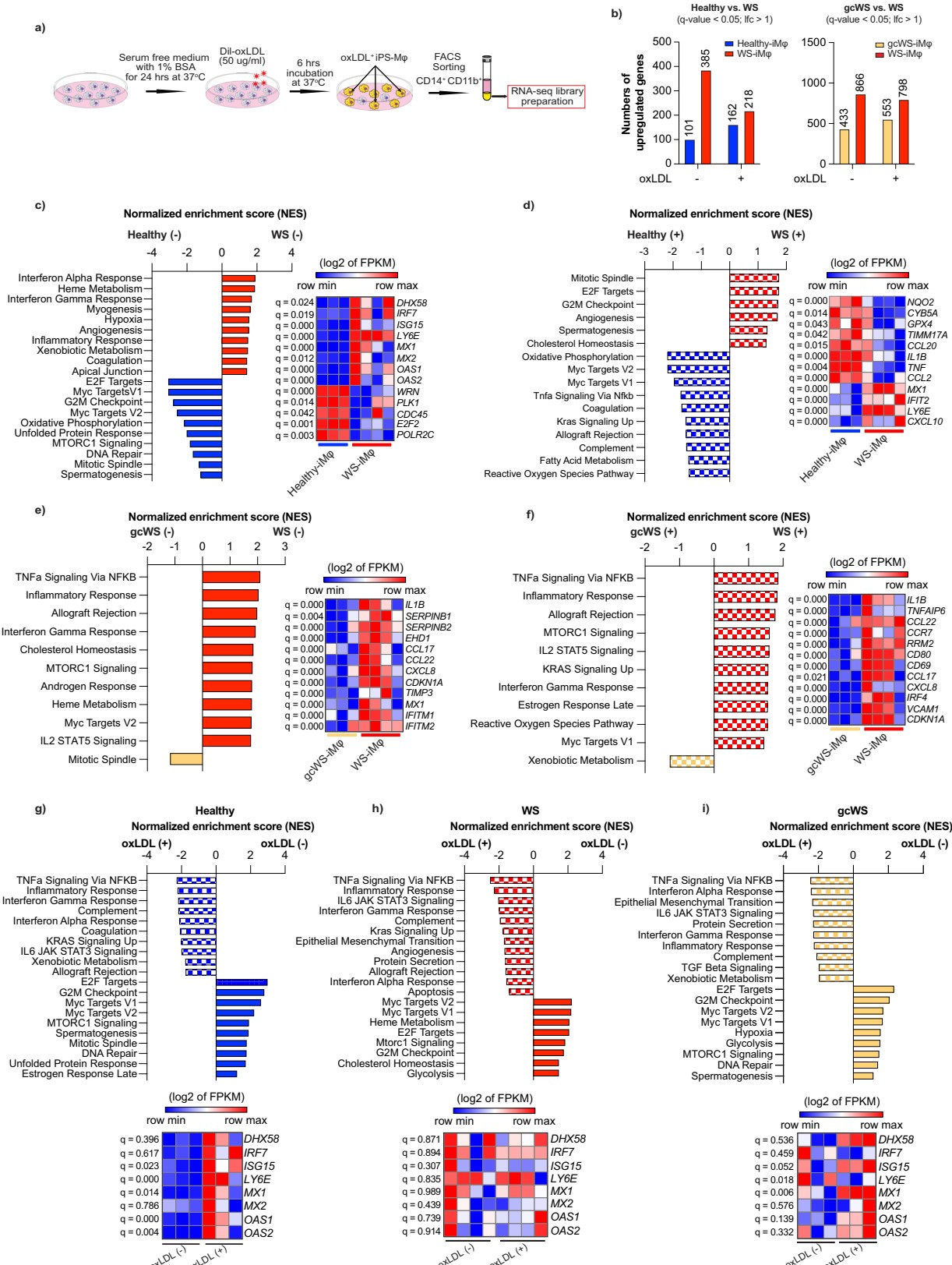

**Fig. 3 | RNA-seq analysis of iMφs. a** Schematic diagram of RNA-seq library preparation. **b** Numbers of DEGs in healthy- (blue), WS- (red), and gcWS-iMφs (yellow) before and after oxLDL treatment. DEGs were derived from Cuffdiff pairwise analysis (q-value < 0.05, lfc >1). Pairwise GSEA of non-treated healthy- vs. WS-iMφs (**c**), oxLDL-treated healthy- vs. WS-iMφs (**d**), non-treated gcWS- vs. WS-iMφs (**e**), oxLDL-treated gcWS- vs. WS-iMφs (**f**), non-treated healthy-iMφs vs. oxLDL-treated healthy-iMφs (**g**), non-treated WS-iMφs vs. oxLDL-treated WS-iMφs (**h**), and non-treated gcWS-iMφs vs. oxLDL-treated gcWS-iMφs (**i**). Bar plots show the top 10 enriched pathways in GSEA, and heatmaps show the enriched gene sets in those top 10 pathways. Heatmaps were derived from the log2 of fragments per kilobase of exon per million reads mapped (FPKM) values. Statistical significance was derived from q-values calculated by Cuffdiff. (−) non-treated; (+) oxLDL-treated. All the experiments in this figure were derived from (n = 3, for healthy-; n = 4 for WS-; and n = 3 for gcWS-iMφ biologically independent samples). oxLDL oxidized low-density lipoprotein.

chromatin using sequencing (ATAC-seq)[45,46]. As oxLDL treatment did not change chromatin accessibility, we performed analyses regardless of oxLDL treatment. In pairwise analysis, we compared differentially accessible regions (DARs) among healthy-, WS-, and gcWS-iMφs. We observed a total of 2531 differentially open chromatin sites between healthy- and WS-iMφs, with only 283 open sites in WS-iMφs (Fig. 4a). Similarly, we observed a total of 1049 differentially open chromatin sites between gcWS- and WS-iMφs, with only 85 open sites in WS-iMφs (Fig. 4b).

We next focused on DARs among healthy- WS-, and gcWS-iMφs (Fig. 4c). K-means clustering showed that only cluster 3 included open chromatin sites in WS-iMφs, which were shared with gcWS-iMφs. All other clusters exhibited open chromatin sites in healthy-iMφs. Also, the sites in clusters 1 and 5 were only gradually opening in gcWS-iMφs (Fig. 4c, d). As a previous study reports a global reduction in chromatin accessibility in senescence-related macular degeneration[47], we speculate that a Mφ-specific reduction in chromatin accessibility in WS-iMφs is associated with cellular senescence.

Next, to identify key molecules, we performed motif enrichment analysis using HOMER. We found that IRF, CEBP, and JUN-AP-1 motifs were enriched in WS-iMφs compared with healthy-iMφs and that ATF, JUN-AP-1, and CEBP motifs were enriched in WS-iMφs compared with gcWS-iMφs. By contrast, the binding sites of several important transcription factors for cellular homeostasis (e.g., GATA, E2F, TCF, and PBX) were closed in WS-iMφs (Fig. 4e, f). Furthermore, to identify representative transcriptional factors in each type of cell, we calculated motif enrichment using the top 10% of open chromatin sites among healthy-, WS-, and gcWS-iMφs and observed that several well-known inflammatory motifs showed higher average enrichment in WS-iMφs compared with healthy-iMφs (Fig. 4g). Intriguingly, enrichment of a few motifs was suppressed in gcWS-iMφs (Fig. 4g). These findings suggest that aging, but not oxLDL treatment, closed many regions that are necessary for proper cellular homeostasis and opened regions that could cause inflammation in WS-iMφs. Unexpectedly, although the inflammatory phenotype was partially rescued in gcWS-iMφs, changes in accessible chromatin regions and reductions in the enrichment of inflammatory motifs were modest (Fig. 4c, d, g).

## Effect of silencing type I IFN signaling on cellular and vascular health in WS-iMφs

To investigate the effects of type I IFN signaling on WS-iMφs, we knocked down *IRF3* and *IRF7* (*IRF3/7*), which are key transcription factors that contribute to the activation of virus-inducible cellular genes, including type I IFN genes, using a lentiviral shRNA system[48]. shRNA-mediated knockdown significantly reduced mRNA levels of *IRF3/7* in healthy-iMφs (Supplementary Fig. 7a), WS-iMφs (Fig. 5a), and gcWS-iMφs (Supplementary Fig. 7h). In accordance with the known induction of transcription of several type I IFN signature genes including *ISG15*, *MX1*, and *MX2* by *IRF3/7*, mRNA levels of *ISG15*, *MX1*, and *MX2* were suppressed after *IRF3/7* knockdown compared with after control shRNA transduction in WS-iMφs (Fig. 5b). Next, we investigated the effects of *IRF3/7* knockdown on cell proliferation and cellular senescence. We found that WS-iMφs exhibited higher proliferation (Fig. 5c) and reduced expression of SA-β-gal (Fig. 5d) and *CDKN2A* mRNA (Fig. 5e) after *IRF3/7* knockdown than after control shRNA transduction. Furthermore, the secretion and mRNA expression of pro-inflammatory cytokines *IL6* and *TNF* (TNFα) were significantly reduced after *IRF3/7* knockdown (Fig. 5f, g). By contrast, these phenomena were not observed in healthy-iMφs (Supplementary Fig. 7b–g) or gcWS-iMφs (Supplementary Fig. 7i–l), perhaps due to their lower expression of IRF3 and 7. Collectively, these results suggest that type I IFN signaling promotes the senescence-associated inflammatory response and, conversely, that silencing the expression of *IRF3/7* suppresses senescence and inflammation, thereby promoting cell proliferation in WS-iMφs.

Moreover, to investigate the effect of *IRF3/7* knockdown on vascular health, we co-cultured vascular cells with *IRF3/7* knockdown iMφs. Interestingly, iMφs-iVEC adhesion was significantly reduced, and reduced cell surface expression of ICAM-1 on WS-iVECs and reduced *ICAM1* mRNA levels were observed in *IRF3/7* knockdown WS-iMφs compared with shCtrl-treated WS-iMφs (Supplementary Fig. 8a–c). Although not statistically significant, there was a trend toward increased contractile marker mRNA expression in WS-iVSMCs upon co-culture with *IRF3/7* knockdown WS-iMφs (Supplementary Fig. 8d). These results suggest that silencing type I IFN signaling in WS-iMφs partially improves vascular health in WS.

## Resurrection of retrotransposons in WS-iMφs

As our culture system did not contain the virus or IFN itself, we speculated that type I IFN signaling is upregulated by a cell-intrinsic mechanism. To identify the trigger of type I IFN signaling in WS-iMφs, we examined factors (i.e., ROS, ER stress, DNA damage, and RTEs) that could potentially induce a type I IFN response other than viral infection[8,16–19]. WS-iMφs showed increased accumulation of cellular ROS and higher mRNA expression of the ROS generator NADPH oxidase 2 (*NOX2*) (Supplementary Fig. 9a, b)[49]. Although *NOX2* inhibition in WS-iMφs significantly reduced ROS generation (Supplementary Fig. 9c), we found no notable association of type I IFN signature genes with ROS in WS-iMφs (Supplementary Fig. 9d).

Next, we investigated whether ER stress initiated the type I IFN response in WS-iMφs by examining the mRNA expression of ER stress-related genes (i.e., *ERN1*, *EIF2AK3*, *HSPA5*, and *ATF6*) but observed no notable changes (Supplementary Fig. 10a). Furthermore, the cGAS-STING signaling pathway, a sensor of cytosolic DNA, was recently identified as a key mediator of the type I IFN response and is implicated in many inflammatory diseases[50]. However, there were no significant differences in the mRNA expression of cGAS (*MB21D1*) and STING (*TMEM173*) genes between healthy- and WS-iMφs (Supplementary Fig. 10b). For further confirmation, we blocked the reverse transcription of RNA to DNA using two nucleoside/nucleotide reverse transcriptase inhibitors, lamivudine and emtricitabine, but observed no notable changes in the expression of *MB21D1*, *TMEM173*, or type I IFN signature genes (Supplementary Fig. 10c–e).

Finally, we analyzed the expression of RTEs in healthy-, WS-, and gcWS-iMφs with or without oxLDL treatment and in PB-derived primary Mφs (PB-Mφs) from healthy age-matched donors and WS patients. To systematically examine global RTE expression in iMφs and PB-Mφs, we used RNA-seq analysis. We identified genomic regions representing RTEs and used the RNA read counts within these regions in each sample to identify differentially expressed RTEs. RNA-seq analysis revealed that iMφs (adjusted p ≤ 0.01; log fold change (lfc) ≥ 1) and PB-Mφs (p ≤ 0.05, lfc ≥0.5) showed 229 and 128 differentially expressed RTEs, respectively (Fig. 6a; Supplementary Fig. 11a). Next, we performed pairwise comparisons of WS-iMφs and gcWS-iMφs with healthy-iMφs to identify upregulated and downregulated RTEs in WS- and gcWS-iMφs. Surprisingly, we found that among 101 upregulated RTEs in gcWS-iMφs, 100 were in common with WS-iMφs (Fig. 6b, c). In addition, WS-derived cells displayed similar upregulation of RTE expression compared with healthy cells (Fig. 6a, d; Supplementary Fig. 11a, b). Interestingly, *WRN* gene correction significantly reduced RTE expression in gcWS-iMφs, specifically in clusters 5, 6, and 7 (Fig. 6a, d). oxLDL treatment did not affect RTE expression in WS-iMφs. However, several RTEs in clusters 1 and 4 were upregulated and RTEs in cluster 2 were downregulated in healthy-iMφs after oxLDL treatment (Fig. 6a, d). There were no notable differences in the proportions of LINE, SINE, and ERV expression between iPS-iMφs and PB-iMφs (Fig. 6e, f; Supplementary Fig. 11c).

RTEs are repressed by several epigenetic modifications, including DNA methylation and H3K9me3 histone modification[22]. To determine

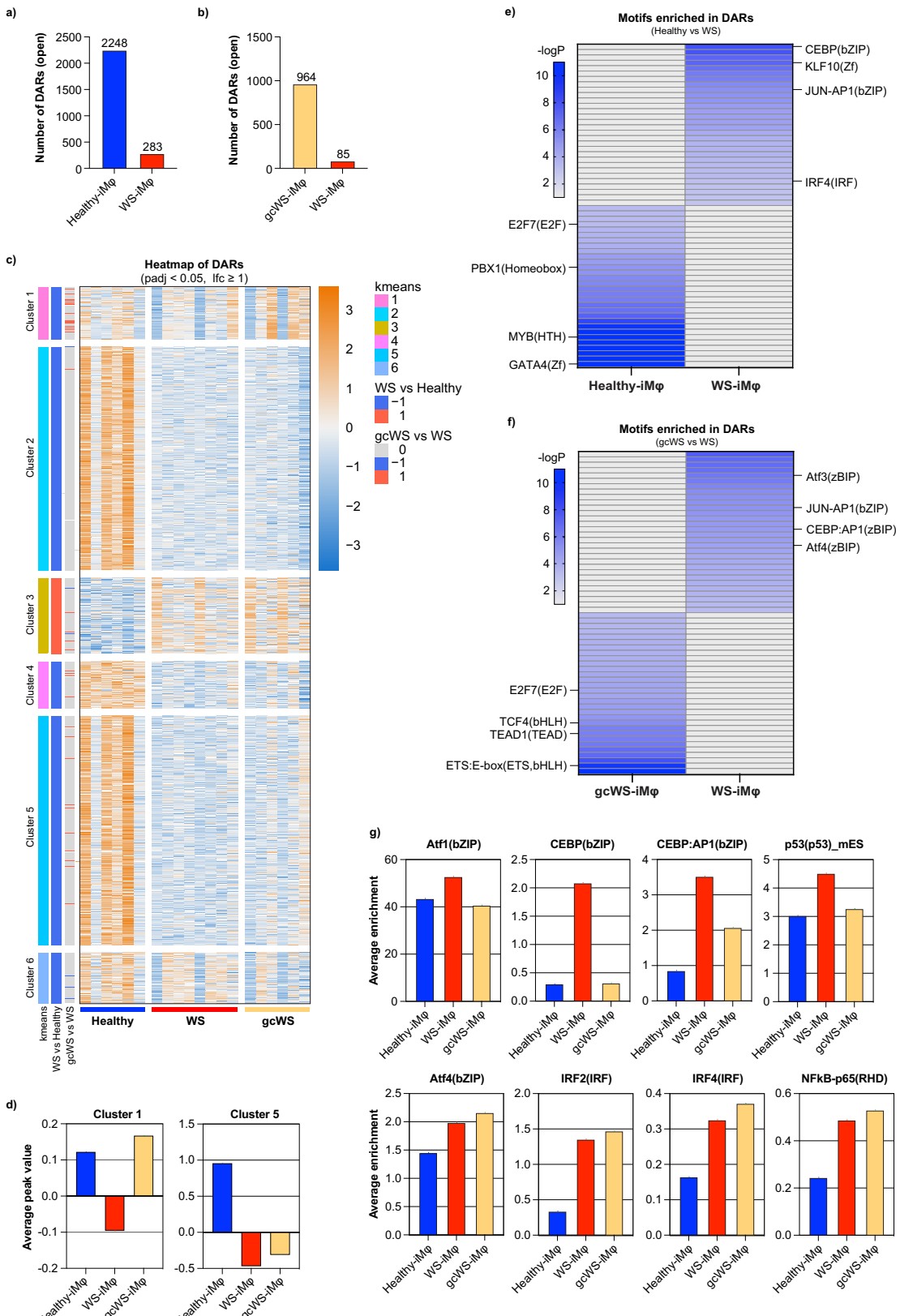

why WS-iMφs cells showed significantly higher levels of RTE expression, we performed H3K9me3 ChIP-seq. RNA-seq pairwise analysis revealed that individual numbers of RTEs, even at the sub-family level, were higher in WS-iMφs than in healthy- and gcWS-iMφs. Consistently, H3K9me3 levels in these regions were significantly lower in WS-iMφs than in healthy-iMφs (Fig. 6e). As a previous study reports that Suv39h-

dependent H3K9me3 chromatin specifically represses intact LINE elements in the mouse ESC epigenome[22], this suggests that loss of H3K9me3 levels de-represses RTEs in WS-iMφ. On the other hand, although we confirmed that the expression of some RTEs was repressed in gcWS-iMφs, there was no significant difference in H3K9me3 levels among groups (Fig. 6a, d, f).

**Fig. 4 | Type I IFN-specific chromatin accessibility profile in WS-iMφs.** Numbers of DARs between healthy- (n = 6) and WS-iMφs (n = 8) (**a**) and between WS- (n = 8) and gcWS-iMφs (n = 6) (**b**). **c** Heatmap of counts per million (CPM) of DARs calculated using DESeq2 in healthy-, WS-, and gcWS-iMφs, with each column representing the CPM of each DAR within a sample and each row representing an individual DAR (adjusted p < 0.05, lfc >1). Differential analysis with DESeq2 (v1.36.0 with default parameters). **d** Clusterwise average peak value in healthy-, WS-, and gcWSiMφs. Heatmaps showing enriched top 30 motifs in DARs by HOMER in healthy- and WS-iMφs (**e**) and in WS- and gcWS-iMφs (**f**); Heatmaps were derived from -logP values of each enriched motifs. -logP values over 10 were defined as dark blue in color. Finding Enriched Motifs in Genomic Regions with HOMER findMotifsGenome.pl with default parameters. **g** Bar plots show average enrichment of individual motifs in healthy-, WS-, and gcWS-iMφs. Finding Enriched Motifs in Genomic Regions with HOMER findMotifsGenome.pl with default parameters. oxLDL oxidized low-density lipoprotein, DAR differtially accessible regions. Source data are provided as a Source Data file.

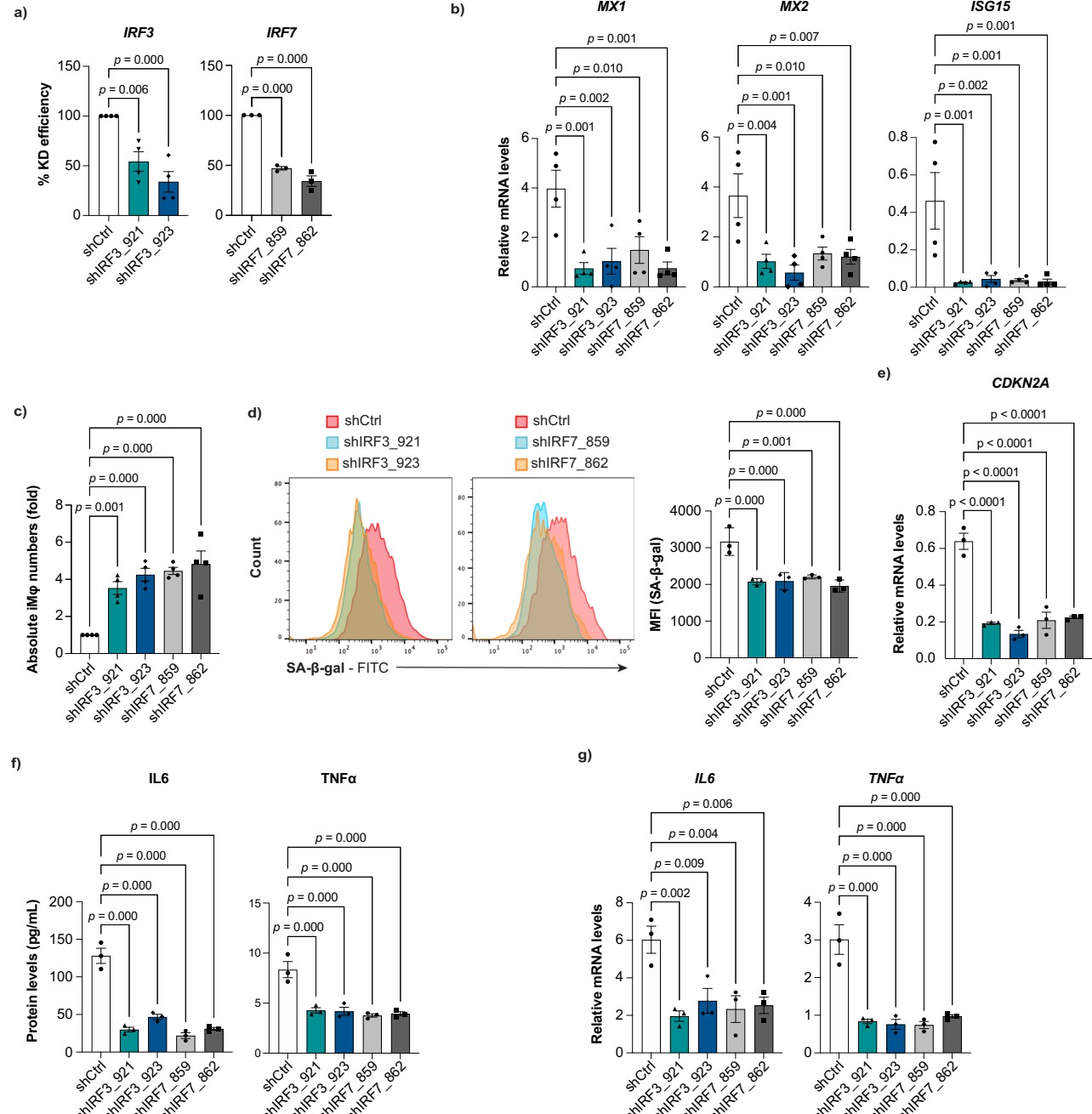

**Fig. 5 | Type I IFN signal-dependent cellular senescence and inflammation in WS-iMφs. a** Percent knockdown (KD) efficiency by shIRF3 (n = 4) and shIRF7 (n = 3). **b** mRNA levels of type I IFN signature genes normalized by *GAPDH* mRNA after lentiviral transduction of shIRF3 and shIRF7 (n = 4). **c** Fold change in absolute numbers of WS-iMφs after lentiviral transduction of shIRF3 and shIRF7 (n = 4). **d** Representative flow cytometric plots of SA-β-gal staining (left) and MFI (right) after lentiviral transduction of shIRF3 and shIRF7 (n = 3).

**e** *CDKN2A* mRNA levels normalized by *GAPDH* mRNA (n = 4). **f** Pro-inflammatory cytokine levels after lentiviral transduction of shIRF3 and shIRF7 (n = 4). **g** *IL6* and *TNFα* mRNA levels normalized by *GAPDH* mRNA after lentiviral transduction of shIRF3 and shIRF7 (n = 4). Data are shown as mean ± SEM of biologically independent samples. One-way ANOVA with Dunnett's multiple comparisons was performed to calculate the p values. Source data are provided as a Source Data file.

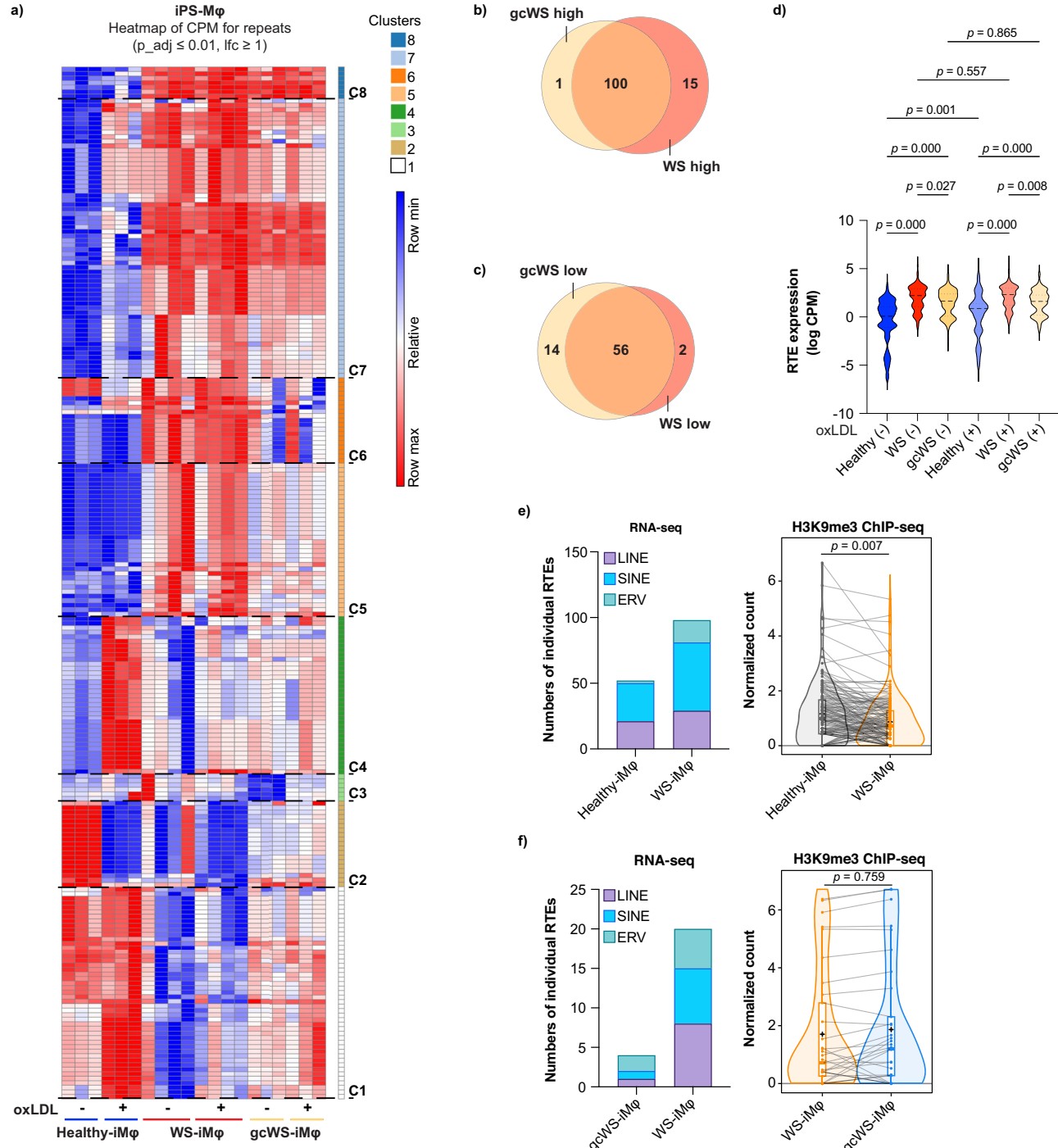

**Fig. 6 | Resurrection of retrotransposons in WS-iMφs. a** Heatmap showing differential expression of RTEs obtained using limma-voom in healthy- (n = 3), WS- (n = 4), and gcWS-iMφs (n = 3) before and after oxLDL treatment, with each column representing a sample group and each row representing an individual RTE. Moderated paired t tests were performed using limma, and the p values were corrected for multiple comparisons using Benjamini Hochberg's method. Venn diagram showing commonly upregulated (**b**) and downregulated (**c**) RTEs in gcWS- (n = 3) and WS-iMφs (n = 4) compared with healthy-iMφs (n = 3) in pairwise analysis (adjusted p < 0.01, lfc >1). Pairwise analysis of individual numbers of RTEs was determined by RNA-seq with moderated paired t tests were performed using limma, and the p values were corrected for multiple comparisons using Benjamini Hochberg's method. **d** Levels of RTE expression healthy- (n = 3), WS- (n = 4), and gcWS-iMφs (n = 3) (adjusted p < 0.01, lfc >1) before and after oxLDL treatment. Data are presented as logCPM values for each group. One-way ANOVA was performed to calculate the p values. **e** Numbers of individual RTEs (adjusted p < 0.01, lfc >1) at sub-family (LINE, SINE, and ERV) levels in non-treated healthy- and WS-iMφs (left),

for which levels of H3K9me3 were determined by ChIP-seq in non-treated healthy- (n = 3) and WS-iMφs (n = 3) (right). Welch Two Sample t test was performed to calculate the p values. Boxes represent the 25–75 percentile ranges with the median of the horizontal line and the mean of the plus. The ends of vertical lines represent the 10 or 90 percentiles. **f** Numbers of individual RTEs (adjusted p < 0.01, lfc >1) at sub-family (LINE, SINE, and ERV) levels in non-treated WS- and gcWS-iMφs (left), for which levels of H3K9me3 were determined by ChIP-seq in non-treated WS- (n = 3) and gcWS-iMφs (n = 3) (right). Two Sample t test was performed to calculate the p values. Boxes represent the 25–75 percentile ranges with the median of the horizontal line and the mean of the plus. The ends of vertical lines represent the 10 or 90 percentiles. oxLDL oxidized low-density lipoprotein, CPM counts per million, C1-8; cluster 1-8, long-interspersed nuclear element (LINE), short-interspersed nuclear element (SINE), and endogenous retrovirus (ERV). In this figure, n represents biologically independent samples. Source data are provided as a Source Data file.

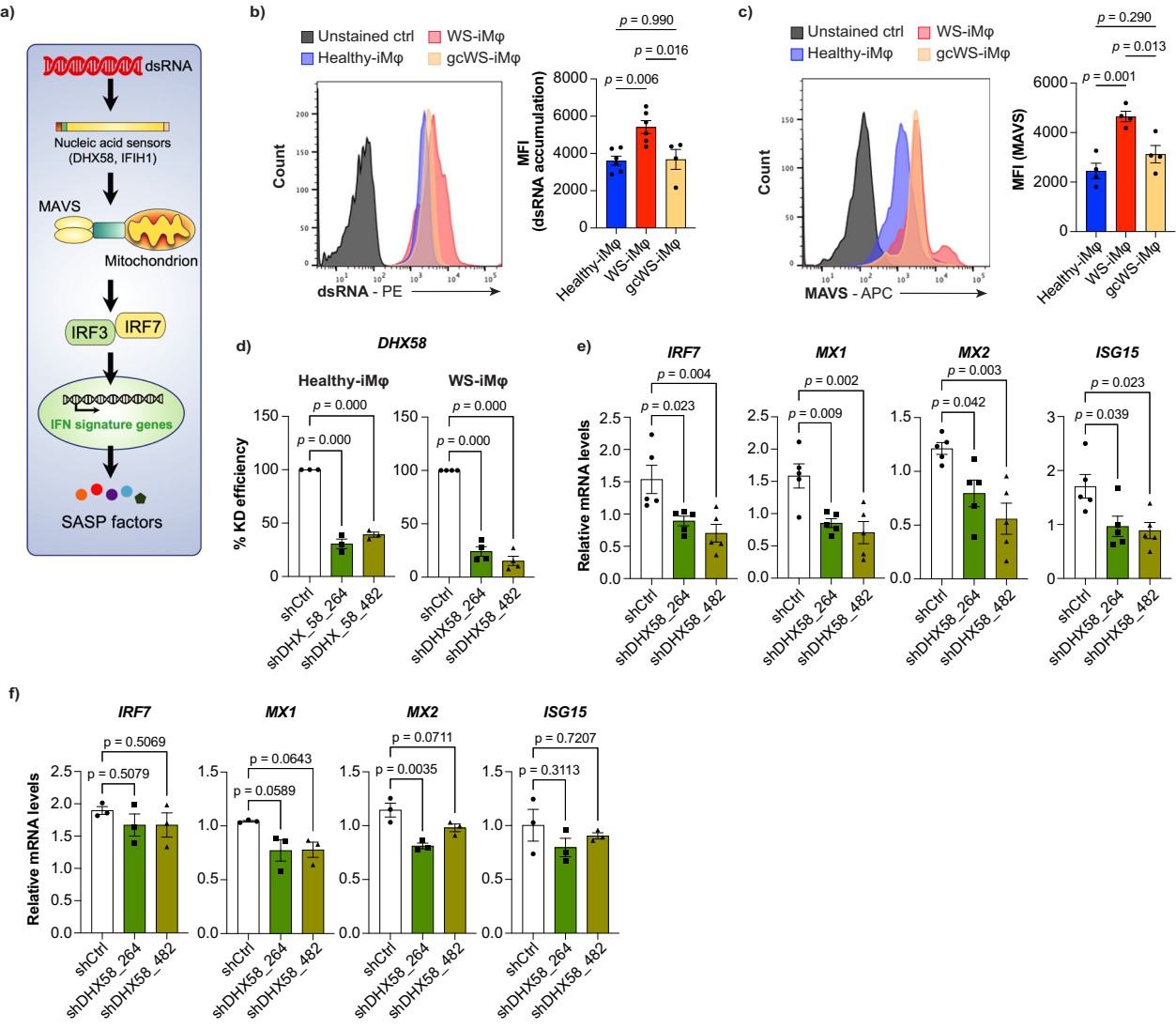

**Fig. 7 | *DHX58*-dependent dsRNA sensing pathway initiates transcription of type I IFN signature genes. a** Schematic diagram of *DHX58*-dependent dsRNA sensing pathway. **b** Representative FACS plot of dsRNA accumulation (left) and MFI of dsRNA accumulation (right) in healthy- (n = 6), WS- (n = 6), and gcWS-iMφs (n = 4), three biologically independent samples over two independent experiments. **c** Representative FACS plot of MAVS expression (left) and MFI of MAVS expression (right) in healthy-, WS-, and gcWS-iMφs (n = 4). **d** Percent knockdown efficiency of lentiviral transduction of shDHX58 in healthy-iMφs (n = 3) (left) and WS-iMφs

(n = 4) (right). mRNA levels of type I IFN signature genes normalized by *GAPDH* mRNA after lentiviral transduction with shDHX58 in WS- iMφs (n = 5) (**e**) and healthy-iMφs (n = 3) (**f**). Data are shown as the mean ± SEM biologically independent. One-way ANOVA with Dunnett's multiple comparisons was performed to calculate the *p* values. MFI mean fluorescent intensity, oxLDL oxidized low-density lipoprotein, MAVS mitochondrial antiviral-signaling protein, IFN interferon. Source data are provided as a Source Data file.

## RTE-induced activation of the innate immune system via *DHX-58* in WS-iMφs

Viral double-stranded (ds)RNA is readily sensed by RIG-I-like receptors and *IFIH1*, leading to propagation of signals to the nucleus via mitochondrial antiviral-signaling protein (MAVS) and, ultimately, initiation of the type I IFN response (Fig. 7a)[51]. RTEs are usually generated in the nucleus and released into the cytoplasm, where they form dsRNA that is detected by cytosolic RNA sensors and evokes the type I IFN response, which is called viral mimicry[52,53]. When we stained cells with anti-dsRNA antibody, we found that only WS-iMφs, but not WS-iPS-derived vascular cells, accumulated more dsRNA than healthy- and gcWS-iMφs (Fig. 7b, Supplementary Fig. 12a, b). We also confirmed that *DHX58* mRNA was upregulated in WS-iMφs (Fig. 3c). Furthermore, flow cytometry showed that MAVS was more highly activated in WS-iMφs than in healthy- and gcWS-iMφs (Fig. 7c). To determine whether dsRNA induces type I IFN signaling through cytosolic RNA sensors, we suppressed a significantly enriched nucleic acid sensor in this pathway,

*DHX58*, in healthy- and WS-iMφs (Fig. 7d). Interestingly, the expression of several type I IFN signature genes, including *IRF7*, *ISG15*, *MX1*, and *MX2*, which are downstream targets of *DHX58*, were significantly downregulated only in WS-iMφs but not healthy-iMφs after knockdown of *DHX58* (Fig. 7e, f). Taken together, these results suggest that RTE-derived dsRNA is detected by *DHX58* and initiates the transcription of type I IFN signature genes in WS-iMφs.

## Discussion
This study describes an efficient technique for investigating vascular aging and human pathologies affecting the cardiovascular system in WS through the direct differentiation of iPSCs into human myeloid and vascular cells. Using these cells, we successfully established an in vitro 2D co-culture system to study atherosclerosis at the molecular level. In WS-iMφs, which are culprits for aberrant inflammation and cellular senescence, transcriptomics (RNA-seq) and open chromatin (ATAC-seq) analysis revealed accelerated activation of type I IFN signaling and

reduced chromatin accessibility of several transcription factors essential for cellular homeostasis regardless of oxLDL treatment, and these phenomena were partially rescued in one allele gene-corrected WS-iMφs. Furthermore, we showed that activation of RTEs led to release of dsRNA into the cytosol, which in turn activated the *DHX58*-dependent nucleic acid sensing pathway followed by type I IFN signaling in WS-iMφs, resulting in aberrant inflammation and cellular senescence. By contrast, gcWS-iMφs showed significantly less accumulated dsRNA, inflammation, and senescence. These results suggest that WS cells themselves become a risk factor for chronic inflammation and atherosclerosis, independent of the high blood glucose or hyperlipidemia frequently observed in WS patients.

Type I IFN signaling was recently recognized as an inducer of atherosclerosis[12–15]. Type I IFN signaling is upregulated by viral infections or exposure to single- or double-stranded nucleic acids via toll-like receptors, cGAS-STING- or RIG-I-like receptor-dependent pathways, and some growth factors and cytokines[54]. Toll-like receptors are cell surface receptors that play crucial roles in identifying extracellular pathogens, including viral RNA[55]. On the other hand, RIG-I-like receptors (e.g., *DDX58*, *DHX58*, and *IFIH1*) detect single-stranded RNA and dsRNA that accumulate in the cytoplasm and evoke type I IFN responses in an *IRF3/7*-dependent manner[52]. Surprisingly, we found upregulation of type I IFN signaling in WS-iMφs even though the culture medium did not contain pathogens or IFN-α/β cytokines. We observed no associations of ROS, ER stress, or DNA sensing via the cGAS-STING pathway with the expression of type I IFN genes. These results suggest that the RTE-RIG-I-like receptor pathway induced type I IFN signaling, leading to cellular senescence and proliferation arrest in WS-iMφs[8,56,57].

The RTE-dependent type I IFN response is well studied in both mouse models and human cells[18,58]. Recent reports suggest that RTE-derived dsRNA activates RIG-I-like receptor-dependent IFN responses in several types of cancer[28,52,59–63]. A recent study demonstrates that reactivation of LINE1 (L1) elements causes cytoplasmic accumulation of L1 cDNA and promotes aging-associated inflammation by activating the type I IFN response. Conversely, inhibiting L1 replication improves the health and lifespan of aged mice[26]. In addition to L1, suppression of SINE Alu elements reverses the senescence phenotype, thereby restoring cell self-renewal properties[28]. Furthermore, ERVs are linked to the induction of growth-inhibiting immune responses. DNA methyltransferase inhibitors upregulate ERVs, triggering cytosolic sensing of dsRNA that results in a type I IFN-dependent immune response and apoptotic cell death[19,29,52]. In various primate and mouse aging models, de-repression of human ERVs provokes cellular senescence and tissue aging[27]. Hence, aberrant RTE expression is associated with inflammation and cellular aging/apoptosis. Moreover, a previous study reports that Suv39h-dependent H3K9me3 chromatin specifically represses intact LINE elements in the mouse ESC epigenome[22]. In addition, *WRN*-null human ESCs show a global loss of H3K9me3 and changes in heterochromatin architecture[64]. Intriguingly, we found that H3K9me3 levels in highly expressed RTE regions were significantly lower in WS-iMφs than in healthy-iMφs. Conversely, in gcWS-iMφs, reactivation of RTE and inflammation pathways, including IFN signature genes, was also subdued, resulting in partial reversal of cellular aging. However, it is noteworthy that H3K9me3 levels were not improved and the change in chromatin-accessible regions was modest in gcWS-iMφs. Previous reports suggest that the expression of LINE itself reduces levels of H3K9me3 by suppressing SUV39H expression[65]. In this case, it is possible that epigenetic changes may be a secondary event. Therefore, the mechanism underlying the increase in RTE in WS remains unclear and warrants further investigation.

In conclusion, this study shows evidence that aberrant RTE expression and upregulation of IFN signaling in Mφs contribute to atherosclerosis development. WS-iMφs in a pro-inflammatory state are the main malefactors that prime the development of and drive atherosclerosis progression in WS patients independent of other risk factors, such as diabetes mellitus and hyperlipidemia. Conversely, *WRN* gene correction or silencing type I IFN signaling attenuated WS-iMφ proliferation, senescence, and inflammation. Taken together, these findings indicate that targeting type I IFN signaling may be an attractive way of preventing and treating atherosclerosis in WS patients.

## Methods
### Cell culture
We recently established healthy-, WS patient-derived, and *WRN* gene-corrected (gc)WS iPSCs[31]. All experiments were performed using three (for healthy and gcWS) or four (for WS) individual iPSC clones obtained from three different age-matched healthy donors and WS patients as well as corresponding gene-corrected iPSCs from WS patients. On every passage, sub-confluent iPSCs were treated with 0.05% trypsin-ETDA (cat. #32778-34, Nacalai Tesque) at 37 °C for 5–6 min and suspended with 1 ml Dulbecco's Modified Eagle's Medium (cat. #08459-64, Nacalai Tesque) with 10% fetal bovine serum (FBS; cat. #FB-1365/500, Biosera) by gentle pipetting. Single cells were plated on newly prepared Matrigel (cat. #356230, Corning)-coated 6-cm plates at 2.5-3 × 10³/cm² with AK02N (ANJINOMOTO) supplemented with 1 μM Y-27632 (cat. #036-24023, Fujifilm). After 24 h, the culture medium was changed to medium without Y-27632. C3H10T1/2 cells were obtained and cultured as previously described[33]. HAECs obtained from Lonza (CC-2535) and human aortic smooth muscle cells obtained from Gibco (C0075C) were cultured according to the manufacturers' protocols. All studies were performed in accordance with the Declaration of Helsinki. The ethics committee of Chiba University, Chiba, Japan (#1145) approved this study.

### Differentiation of Mφs from iPSCs
We previously demonstrated that multipotent HPCs can be efficiently generated from human ESCs and iPSCs[32–35]. Here, we modified the PS-sac method to generate functional Mφs (Supplementary Fig. 1a). To obtain HPCs, PS-sac-like structures in culture were maintained for 2 weeks to obtain CD34⁺ CD43⁺ cells, followed by an additional 7 days of culture with a cytokine cocktail supported by growth-arrested C3H10T1/2 feeder cells. Briefly, iPSCs were dissociated into small colonies (<100 cells) with the aid of CTK (phosphate buffered saline (PBS) containing 0.25% trypsin (cat. #15090-046, Gibco), 1 mM CaCl₂, and 20% knock-out serum replacement (KSR; cat. #10828028, Gibco)) and cultured on a C3H10T1/2 feeder layer in differentiation medium (Iscove modified Dulbecco medium (IMDM) supplemented with a cocktail of 10 μg/mL human insulin, 5.5 μg/mL human transferrin, 5 ng/mL sodium selenite, 2 mM L-glutamine, 0.45 mM-monothioglycerol, 50 g/mL ascorbic acid, 15% highly filtered FBS) supplemented with 20 ng/mL recombinant human VEGF (cat. #293-VE, R&D Systems), and 10 ng/mL human BMP4 (cat. #314-BP/CF-500, R&D Systems) in a low O₂ (5%) incubator at 37 °C. On day 4, the differentiation medium was changed to medium supplemented with 20 ng/mL recombinant human VEGF, 50 ng/mL human bFGF (cat. #064-04541, Fujifilm), 10 μM SB431542 (cat. #192-16541, Fujifilm), and 10 U/mL heparin (cat. #873334, Awai Pharma Co., Ltd.) and kept under low O₂. On day 7, the differentiation medium was changed to a medium without human bFGF and SB431542 (but including the other two factors at the same concentrations) and cultured in an atmospheric (20%) O₂ incubator. Finally, on day 10, the medium was changed to medium without heparin (but including recombinant human VEGF at the same concentration). On days 14-15, PS-sacs were trypsinized with 2.5% trypsin-EDTA (cat. #209-16941, Fujifilm), gently crushed by pipetting, and passed through a 40-μm cell strainer. Cells were then spun down, resuspended in staining medium (2% FBS in PBS), and stained with anti-CD34 (cat. #343514, Biolegend, 1:100 dilution) and anti-CD43 (cat. #343206, Biolegend, 1:100 dilution) antibodies. FACS-sorted CD34⁺

CD43$^+$ HPCs were cultured on a fresh C3H10T1/2 feeder layer in 24- or 96-well culture plates at a density of $1 \times 10^5$ cells/well for 24-well plates and $5 \times 10^4$ cells/well for 96-well plates and maintained in differentiation medium supplemented with 25 ng/mL SCF, 5 ng/mL TPO, 50 ng/mL M-CSF, 50 ng/mL GM-CSF, 25 ng/mL G-CSF, and 25 ng/mL IL-3 (all from R&D Systems).

### Differentiation of vascular cells from iPSCs

iPSC-derived vascular cells were induced using a modified version of a previously established protocol[66,67]. To generate VPCs, we took advantage of the previously established embryonic stem-sac method[32–35]. iPSCs were dissociated into small colonies (<100 cells) with the aid of CTK (PBS containing 0.25% trypsin, 1 mM CaCl$_2$, and 20% KSR) and cultured on a C3H10T1/2 feeder layer in differentiation medium supplemented with 20 ng/mL recombinant human VEGF in an incubator at 37 °C. The culture medium was changed with fresh medium every 3 days. On days 10-11, PS-sacs were trypsinized with 2.5% trypsin-EDTA, gently crushed by pipetting, and passed through a 40-μm cell strainer. Cells were then spun down, resuspended in staining medium, and stained with anti-CD34 (cat. #343508, Biolegend, 1:100 dilution), anti-VEGF-R2 (CD309; cat. #359912, Biolegend, 1:100 dilution), and anti-TRA 1-60 (cat. #560173, BD Pharmingen, 1:50 dilution) antibodies. FACS-sorted cells were seeded (CD34$^+$ cells for VEC induction and CD34$^-$ VEGF-R2$^+$ cells for VSMC induction) on a C3H10T1/2 feeder layer at a density of $1.5 \times 10^4$ cells/cm$^2$ on a 24-well plate and maintained in differentiation medium supplemented with 10% FBS with (for VECs) or without (for VSMCs) 100 ng/mL recombinant human VEGF.

### Differentiation of monocytes/macrophages from human PB-derived mononuclear cells

PB from healthy donors and WS patients was obtained and purified to mononuclear cells according to our previous study[31]. PB-derived CD34$^-$ mononuclear cells were then stained with anti-CD33 (1:100 dilution), anti-CD43 (1:100 dilution), anti-CD14 (1:100 dilution), and anti-CD11b (1:100 dilution) antibodies followed by incubation for 30 min on ice in the dark, washed with cold staining medium, and sorted into CD33$^+$CD43$^+$CD14$^+$CD11b$^+$ monocytes/macrophages using a BD FACSAria IIIu system (Becton Dickinson Japan, Tokyo, Japan).

### Cell sorting and flow cytometric analyses

Cells were suspended in staining medium, incubated for 30 min with appropriate antibodies on ice in the dark, washed with cold staining medium, and sorted or analyzed using a BD FACSAria IIIu system (Becton Dickinson Japan, Tokyo, Japan) or BD FACS Canto II cytometer (Becton Dickinson Japan, Tokyo, Japan).

### Giemsa staining

On day 21 of differentiation, iMφs were stained with Hemacolor (cat. #111661, Merec) according to the manufacturer's protocol. Stained cells were air-dried and observed using a microscope (Nikon Eclipse Ts2R, Japan), and image acquisition was performed using a Nikon camera and NIS-Elements L software (version- 1.00).

### RT-qPCR

Total RNA was extracted using an RNeasy Micro Kit (cat. #74034, Qiagen, Hilden, Germany) and reverse-transcribed using SuperScript VILO™ Master Mix (cat. #11755250, Thermo Fisher Scientific, Waltham, MA). Primer sets were designed by our laboratory or purchased from Thermo Fisher Scientific. Quantitative real-time PCR was carried out with SYBR™ Green PCR Master Mix (cat. #4472908, Applied Biosystems) using a Bio-Rad CFX real-time PCR system. Gene expression was analyzed relative to that of *GAPDH*. All the oligonucleotide primers used in this study were listed in Supplementary Data 2.

### Immunocytochemistry

For the detection of intracellular proteins, vascular cells were cultured in four-well slide chambers (cat. #SCS-N04, Matsunami). On days 17-18 of differentiation, VECs or VSMCs were washed twice with PBS and fixed with 4% paraformaldehyde (cat. #161-20141, Fujifilm) in PBS for 10 min at room temperature. Cells were then washed three times with PBS and permeabilized with 0.1% Triton X100 (cat. #35501-15, Nacalai Tesque) and 10% FBS (Biosera) in PBS for 30-40 min, followed by blocking for at least 1 h at room temperature with Blocking One (cat. #03953-66, Nacalai Tesque). Blocked cells were incubated with primary antibodies diluted in blocking solution at 4 °C overnight on a rotator. On the following day, after three washes with PBS, appropriate fluorescence-labeled secondary antibodies were added and incubated at room temperature for at least 1 h in the dark, after which cells were washed thoroughly with PBS and mounted on microscope slides using ProLong™ Diamond Antifade Mountant with DAPI (cat. #P36962, Invitrogen, Carlsbad, CA). Slides were allowed to air-dry before examination using fluorescence microscopy (Nikon Eclipse Ts2R, Japan), image acquisition was performed using a Nikon camera and NIS-Elements L software (version- 1.00), and images were merged using Adobe Photoshop software. Primary antibodies were anti-α-SMA (1:1,000, cat. #ab7817, Abcam, Cambridge, UK), anti-calponin-1 (1:500, cat. #abt129, Merck-Millipore, Darmstadt, Germany), and anti-VE cadherin (1:500, cat. #ab33168, Abcam). Secondary antibodies were goat anti-mouse IgG H&L (1:200, cat. #ab150113, Abcam) and goat anti-rabbit IgG H&L (1:200, cat. #ab150078, Abcam).

### oxLDL and acetylated LDL uptake assay

iMφs were serum-starved for 24−36 h in IMDM supplemented with 1% bovine serum albumin (BSA) on day 20 of differentiation. oxLDL labeled with 1, 1′-dioctadecyl-3,3′,3′- tetramethylindocarbocyanine (DiI; cat. #L34358, Invitrogen) was added to cells at a concentration of 50 μg/mL with IMDM containing 1% BSA and incubated for 5−6 h in a 37 °C incubator.

iVECs and HAECs were serum-starved for 24−36 h in IMDM supplemented with 1% BSA on day 16 of differentiation (in the case of iVECs). Acetylated LDL labeled with DiI (cat. #BT-902, Alfa Aesar) was added to cells at a concentration of 50 μg/mL with IMDM containing 1% BSA and incubated for 5−6 h in a 37 °C incubator.

### Apoptosis assay by flow cytometry

On day 20 of differentiation, healthy-, WS-, and gcWS-iMφs were serum-starved for 24−36 h, incubated for 5−6 h with DiI-oxLDL at a concentration of 50 μg/mL, and washed three times with PBS. Apoptosis assay was performed for untreated and oxLDL-treated iMφs using the FITC Annexin V detection kit (cat. #556547, BD Pharmingen) according to the manufacturer's protocol and analyzed with a FACS CantoII cytometer (BD).

### Senescence assay by flow cytometry

On day 20 of differentiation, healthy-, WS-, and gcWS-iMφs were serum-starved for 24−36 h, incubated for 5−6 h with DiI-oxLDL at a concentration of 50 μg/mL, and washed three times with PBS. Senescent SA-β-gal$^+$ cells were then detected among untreated and oxLDL-treated iMφs using the Cellular Senescence Flow Cytometry Assay Kit (cat. #CBA-232, Cell Biolabs, Inc.) according to the manufacturer's protocol and analyzed with a FACS CantoII cytometer (BD).

### γH2AX staining

On day 21 of differentiation, $5 \times 10^4$ healthy-, WS-, and gcWS-iMφs were washed in PBS and fixed with 4% paraformaldehyde for 10 min at room temperature. Cells were then permeabilized with permeabilization solution (PBS containing 0.1% Triton X-100) for 5 min at room temperature followed by washing with PBS for 5 min three times. Cells were blocked with blocking solution (0.3% TritonX-100 and 3% BSA in

PBS) for 1 h at room temperature. Cells were incubated with phospho-histone H2A.X (Ser139) (20E3) rabbit mAb (cat. #9718, Cell Signaling) diluted 1:200 in blocking buffer at 4 °C overnight on a rotator. The next day, cells were washed three times with PBT for 5 min each, followed by staining with secondary antibody goat anti-rabbit IgG H&L (1:200, cat. #ab150078, Abcam) for 1 h at room temperature in the dark, after which cells were washed thoroughly with PBS and analyzed with a BD FACS Canto II cytometer. Cells were mounted on microscope slides using ProLong™ Diamond Antifade Mountant with DAPI (cat. #P36962, Invitrogen, Carlsbad, CA). Slides were allowed to air-dry before examination using fluorescence microscopy (BZ-X700, Keyence), and images were merged using Adobe Photoshop software.

### Coculture of iMφs with iPSC-induced vascular cells
On day 20 of differentiation, iMφs were serum-starved for 24−36 h and incubated for 5−6 h with DiI-oxLDL at a concentration of 50 µg/mL. After washing three times with PBS, untreated or oxLDL-treated $7 \times 10^4$ iMφs (on a 24-well plate) were seeded onto iPSC-derived mature vascular cells (on day 17) and maintained with iMφ differentiation medium for 72 h until further analysis. Cell culture supernatant was collected and filtered with a 0.22-µm filter followed by ELISA to assess protein levels. Cells were then trypsinized with 0.05% trypsin-EDTA (Gibco), gently crushed by pipetting, spun down and resuspended in staining medium, and stained with anti-CD14 antibody (cat. #A22331, Backman Counter, 1:100 dilutions) for 30 min on ice in the dark. CD14⁻ iPSC-derived vascular cells were then sorted for RT-qPCR gene expression analysis.

### Mφ-VEC adhesion assay
On day 20 of differentiation, iMφs were serum-starved for 24−36 h and incubated for 6 h with DiI-oxLDL at a concentration of 50 µg/mL. After washing three times with PBS, untreated or oxLDL-treated $7 \times 10^4$ iMφs were seeded onto iPSC-derived mature vascular cells on day 17 and maintained with iMφ differentiation medium for 72 h until further analysis. Cells were gently washed with PBS three times, trypsinized with 0.05% trypsin-EDTA (Gibco), gently crushed by pipetting, spun down and resuspended in staining medium, and stained with anti-CD14 antibody for 30 min on ice in the dark. CD14⁺ Mφs were then counted with CountBright (cat. #C36950, Invitrogen) using a BD FACSAria IIIu system.

### RNA-seq and analysis
On day 20 of differentiation, iMφs were serum-starved for 24−36 h and incubated for 6 h with DiI-oxLDL at a concentration of 50 µg/mL. Cells were then washed three times with PBS, resuspended in staining medium, incubated for 30 min with anti-CD14 and anti-CD11b antibodies (cat. #301318, Biolegend, 1:100 dilutions) on ice, and washed with cold staining medium. Then, oxLDL⁺ CD14⁺ CD11b⁺ cells were sorted into Buffer RLT Plus. Total RNA from untreated or oxLDL-treated healthy-, WS-, and gcWS-iMφs was isolated using an RNeasy Micro Kit (QIAGEN). RNA-seq libraries were prepared from at least three biological replicates according to the manufacturer's protocol. Briefly, -10 ng total RNA was used as input for cDNA conversion using a SMART-Seq v4 Ultra Low Input RNA Kit for Sequencing (cat. #634890, Clontech, Takara). cDNA was fragmented using an S220 Focused-ultrasonicator (Covaris). The cDNA library was then amplified using a NEBNext® Ultra™ DNA Library Prep Kit for Illumina (cat. #E7370L, New England Biolabs). Finally, NEBnext library size was estimated using a bioanalyzer with an Agilent High Sensitivity DNA kit. Sequencing was performed using a NextSeq500 System (Illumina) with a single-read sequencing length of 60 bp. TopHat (version 2.1.1; with default parameters) was used to map to the reference genome (UCSC/hg19) with annotation data from iGenomes (Illumina). Levels of gene expression were quantified using Cuffdiff (Cufflinks version 2.2.1; with default parameters).

### ATAC-seq and analysis
On day 21 of differentiation, untreated or oxLDL-treated healthy-, WS-, and gcWS-iMφs were stained with anti-CD14 and anti-CD11b antibodies, incubated for 30 min on ice, and washed and sorted into $5 \times 10^3$ CD14⁺ CD11b⁺ cells in PBS containing 2% FBS. Library preparation for ATAC-seq was performed cells with a Nextera DNA Sample Preparation kit (cat. #FC-121-1030, Illumina) according to a previously reported protocol[45,46]. Libraries for ATAC were sequenced with a NextSeq500 System (Illumina) to generate single-end 60-bp reads. Sequences were aligned to human genome sequences (hg19) using Bowtie2 (default setting)[68]. Briefly, duplicate, unmapped, or poor-quality reads, mitochondrial reads, and overlaps with the ENCODE blacklist were removed. Mapped reads were subsampled using samtools to make the numbers of reads in all samples the same. Macs2 (version 2.2.6) was used to call peaks using nomodel, a narrow peak option. Using a q-value cutoff of 0.01, accessible peaks were detected in each sample. The catalog of all peaks called in any samples was produced by merging all called peaks that overlapped by, at least, one base pair using the Bedtools merge function. The Bedtools map function was used to count the reads at each region in the catalog using bed files of each sample. The read count matrix of each sample was used for the detection of differentially accessible regions (DARs) by using DESeq2. Normalized read counts obtained using DESeq2 for the heatmap were z-score-scaled and plotted. For motif analysis, findMotifsGenome.pl of Homer was used with the -size 200 -mask option.

### RTE enrichment analysis
Genomic coordinates of RTEs, along with their names, classes, and families, were downloaded from the UCSC Genome Browser using the Table Browser with group set to Repeats and track set to RepeatMasker. Each RTE was assigned reads obtained from the mapped RNA-seq samples within its genomic coordinates. Differentially expressed RTEs between sample groups were calculated using limma-voom[69].

### Detection of total cellular ROS
On day 20 of differentiation, iMφs were serum-starved for 24−36 h and incubated for 6 h with 50 µg/mL DiI-oxLDL. Total cellular ROS was detected in untreated and oxLDL-treated iMφs with the CellROX Deep Red Flow Cytometry Assay Kit (cat. #C10491, Invitrogen) according to the manufacturer's protocol and analyzed with a FACS AriaIII cytometer (BD).

### Knockdown assay
Oligonucleotides of shRNA against *IRF3*, *IRF7*, *DHX58*, and *LacZ* (control) were inserted into CS-CDFRfa-EPR lentiviral vector plasmid DNA[48]. The inserted oligonucleotide sequences were as follows:

shCtrl,
CCGGTGTTGGCTTACGGCGGTGATTTCTCGAGAAATCACCGCCG
TAAGCCAACTTTTTG
shIRF3_921:
GATCCCCGCCAACCTGGAAGAGGAATTTCTCGAGAAATTCCTCT
TCCAGGTTGGCTTTTTGGAAAT
shIRF3_923:
GATCCCCGATCTGATTACCTTCACGGAACTCGAGTTCCGTGAAG
GTAATCAGATCTTTTTGGAAAT
shIRF7_859:
GATCCCCGCTGGACGTGACCATCATGTACTCGAGTACATGATGG
TCACGTCCAGCTTTTTGGAAAT
shIRF7_862:
GATCCCCCGCAGCGTGAGGGTGTGTCTTCTCGAGAAGACACACC
CTCACGCTGCGTTTTTGGAAAT
shDHX58_264:
GATCCCCCTGTTCGATGACCGCAAGAATCTCGAGATTCTTGCGG
TCATCGAACAGTTTTTGGAAAT
shDHX58_482:

GATCCCCGCCAGTACCTAGAACTTAAACCTCGAGGTTTAAGTTC
TAGGTACTGGCTTTTTTTGGAAAT

Obtained plasmid DNA was co-transfected with pMD2.G and psPAX2 (both from Addgene) into HEK293T cells using the CalPhos™ Mammalian Transfection Kit (Clontech). HEK293T cells were maintained in IMDM (Gibco) supplemented with 10% fetal calf serum and pen strep L-glutamine (PSG, cat. #10378-016, Gibco). After a medium change with IMDM (10% FBS, PSG, and 1 mM sodium butyrate (B5887, Sigma Aldrich)) on the next day, lentiviruses were concentrated using himac High-Speed Refrigerated Centrifuge (Hitachi CR 21GIII, rotor model - R18A, 42,200 × g, 4.5 h, 4°C) and titrated on MOLM-13 AML cell lines.

## CUT&TAG of histone modifications

On day 21 of differentiation, healthy-, WS-, and gcWS-iMφs were stained with anti-CD14 and anti-CD11b antibodies, incubated for 30 min on ice, and washed and sorted into $2 \times 10^4$ CD14$^+$ CD11b$^+$ cells in PBS containing 2% FBS. Libraries were prepared using a CUT&Tag-IT™ Assay Kit (Active motif) according to the manufacturer's protocol. Histone H3K9me3 monoclonal antibody (#61013, Active motif, 1:50 dilutions) was used for the reaction. Sequencing was performed using NextSeq500 (Illumina) with a single-read sequencing length of 60 bp. Sequences were aligned to human genome sequences (hg19) using Bowtie2 (default setting)[68]. Briefly, duplicate, unmapped, or poor-quality reads, mitochondrial reads, and overlaps with the ENCODE blacklist were removed. Mapped reads were subsampled using samtools to make the numbers of reads in all samples the same. The Bedtools map function was used to count the reads at selected RTE regions using bed files of each sample. The read count matrix of each sample was used for the normalization by using DESeq2.

## dsRNA analysis by J2 staining

On day 21 of differentiation, $5 \times 10^4$ healthy-, WS-, and gcWS-iMφs were washed in PBS and fixed with 4% paraformaldehyde for 10 min at room temperature. Cells were then permeabilized with permeabilization solution (PBS containing 0.1% Triton X-100) for 30 min at room temperature, after which cells were incubated with anti-dsRNA monoclonal antibody J2 (cat. #RNT-SCI-10010200, Jena Bioscience) diluted 1:40 in permeabilization solution for 30 min on ice. Following a wash with permeabilization solution, cells were stained with secondary goat anti-rabbit IgG H&L (cat. #ab150078, Abcam) diluted 1:200 in permeabilization solution for 30 min on ice in the dark. After incubation, cells were washed twice in permeabilization solution and analyzed with a BD FACS Canto II cytometer.

## MAVS staining

On day 21 of differentiation, untreated or oxLDL-treated $5 \times 10^4$ healthy-, WS-, and gcWS-iMφs were washed in PBS and fixed with 4% paraformaldehyde for 10 min at room temperature. Cells were then permeabilized with a permeabilization solution for 30 min at room temperature. Cells were resuspended in permeabilization solution, and MAVS monoclonal antibody (clone: ABM28H9) conjugated with APC (cat. #17-9835-41, eBioscience, 1:50 dilutions) was added to the cells followed by 30 min incubation on ice in the dark. After incubation, cells were washed twice in permeabilization solution and analyzed with a BD FACS Canto II cytometer.

## Statistical analysis

All analyses were performed with three biological replicates with at least n = 3. Data are shown as the mean ± standard error of the mean (SEM). One-way and two-way ANOVAs with Tukey's and Dunnetts's multiple comparison tests were performed to determine statistical significance using GraphPad Prism 10.2.1 (GraphPad Software, Inc., La Jolla, CA, USA). No statistical method was used to predetermine the sample size. No data were excluded from the analyses.

## Reporting summary

Further information on research design is available in the Nature Portfolio Reporting Summary linked to this article.

## Data availability

The NGS data generated in this study were deposited in the NCBI Gene Expression Omnibus (GEO) as a SuperSeries under accession code GSE247722 [https://www.ncbi.nlm.nih.gov/geo/query/acc.cgi?acc=GSE247722] composed of the following SubSeries: GSE247717 (iPS-macrophage RNA-seq), GSE247721 (RTE enrichment analysis of iPS-macrophages and PB-macrophages), GSE247710 (iPS-VEC RNA-seq), GSE247718 (iPS-VSMC RNA-seq), GSE247716 (ATAC-seq), and GSE247705 (CUT&TAG ChIP-seq). The authors declare that all data supporting the findings of this study are available within the article and its Supplementary Information file. Source data are provided with this paper.

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

## Acknowledgements

This research was supported by the Japanese Government MEXT (Monbukagakusho) scholarship (S.K.P.), the Tokyo Biochemical Research Foundation (N.T.), the JSPS KAKENHI Grant Number 23K18284 (N.T.), the Japan Agency for Medical Research and Development under Grant Numbers JP21bm0804016 (Y.M., K.Y., and K.E.), JP21jm0210096 (K.Y.), JP22ym0126066 (K.Y.), JP23ek0109622 (K.Y.), and the Ministry of Health, Labor and Welfare of Japan under Grant Number JPMH21FC1016 (K.Y.).

## Author contributions

S.K.P. and N.T. designed the study and analyzed the data. S.K.P. carried out the laboratory experiments and created the figures. M.S., Y.M., H. Kato., Y.O., H. Kaneko., and N.T. established the iPSCs. S.K.P., M.S., and N.T. established the iPSC-derived Mφs and vascular cells. S.K.P. and N.T. prepared the RNA-seq, ATAC-seq, and CUT&TAG ChIP-seq libraries. M.F., B.R., and A.K. sequenced the RNA-seq, ATAC-seq, and CUT&TAG ChIP-seq libraries. M. Oshima and N.T. performed the bioinformatic analysis for RNA-seq, ATAC-seq, and CUT&TAG ChIP-seq experiments. A.P. performed the bioinformatic analysis for RTE enrichment experiments. M. N. provided the Sendai virus vectors. K.T. generated lentiviruses for knock-down assay. S.K.P and N.T. wrote the manuscript. A.I., K.Y., K.E., and N.T. discussed the data and critically reviewed the manuscript. K.Y., K.E., and N.T. supervised the study. All authors approved the final version of the manuscript.

## Competing interests

The authors declare no competing interests.

## Additional information

[1]Department of Regenerative Medicine, Graduate School of Medicine, Chiba University, Chiba, Japan. [2]Division of Stem Cell and Molecular Medicine, Center for Stem Cell Biology and Regenerative Medicine, The Institute of Medical Science, The University of Tokyo, Tokyo, Japan. [3]Combinatics Inc., Chiba, Japan. [4]Department of Endocrinology, Hematology and Gerontology, Graduate School of Medicine, Chiba University, Chiba, Japan. [5]Department of Molecular Oncology, Graduate School of Medicine, Chiba University, Chiba, Japan. [6]Gene Expression Laboratory, Salk Institute for Biological Studies, La Jolla, CA, USA. [7]TOKIWA-Bio, Inc., Tsukuba, Japan. [8]Department of Clinical Application, Center for iPS Cell Research and Application, Kyoto University, Kyoto, Japan. [9]Present address: Hibernation Metabolism, Physiology and Development Group, Institute of Low Temperature Science, Hokkaido University, Sapporo, Japan. ✉e-mail: kyokote@faculty.chiba-u.jp; kojieto@cira.kyoto-u.ac.jp; tnaoya19760517@gmail.com

