## [Peer Review File · Nature Communications]

REVIEWER COMMENTS

Reviewer #1 (Remarks to the Author):

This study presents some potentially novel observations about the mechanism by which macrophages in Werner syndrome may influence atherosclerosis. The authors use coculture of iPSC-derived macrophages and either vascular smooth muscle cells (iVSMCs) or vascular endothelial cells (iVECs) to demonstrate potential differences in the function of healthy and Werner syndrome (WS) macrophages that might be relevant in atherosclerosis. The most intriguing result is that the WS macrophages may have an activated response to type I IFN-related genes due to retrotransposon expression. As noted in major comments, the effect sizes are modest in a number of cases and may depend on the presence of oxidized LDL (oxLDL), so additional studies are needed to validate the study and increase the overall significance¹.

Major Comments

1. While the first two paragraphs of the introduction are accurate, they should focus on Werner syndrome and its pathology since that is the focus of this study. Also, in the Introduction, provide evidence for the, address the role of type I interferons in atherosclerosis.
2. Assay for differences in monocyte receptor adhesion molecule VCAM-1 on Healthy- and WS-iVEC to address whether this explains difference in macrophage adhesion.
3. Except for TNF-alpha mRNA and protein expression, the mRNA expression of cytokines in coculture with iVECs appears to depend on the presence of oxLDL. Performing the experiment with healthy iVECs and WS macrophages could identify a specific WS-macrophage effect. Further, since there are two types of macrophages and responses are measured with and without oxLDL, a two factor ANOVA should be performed. An interaction effect should arise from the two-factor ANOVA.
4. Based on mRNA and contractile protein expression, reduced contractile protein demonstrated, but no evidence is provided that iVSMCs are synthetic. (lines 169-170).
5. iVSMC proliferation and reduced gene expression of SMC contractile proteins depends on oxLDL with WS-macrophages. This appears to be due to an interaction effect of the WS-macrophages and ox-LDL. At the least, these statements need to be qualified (lines 162-163 and Fig. 2c).
6. With respect to the upregulation of endogenous type I IFN signals in WS-macrophages relative to healthy-macrophages (Lines 190-195, Fig 3g), there is some suppression due to oxLDL and it is unclear that in the presence of oxLDL that there is a difference between the healthy and WS macrophages. Statistical comparison of these conditions is needed to validate the claim. In contrast, apoptosis genes are increased in WS-macrophages after addition of oxLDL relative to healthy case.
7. To show that the knockdown of IRF3/7 does not affect type I IFN gene signatures, senescence, and cytokines expression is specific to WS-macrophages, similar knockdown experiments need to be performed with healthy macrophages.

8. Lines 210-215. Show level in WS-macrophages relative to healthy macrophages of enrichment of IRF and other sites noted

9. Likewise to support the specificity of the knockdown of DHX58 on IRF7, ISG15, MX1, and MX2 in WS-macrophages, perform similar experiments with healthy macrophages (Lines 280-285 and Fig. 7).

Specific Comments.

1. Extended Data Fig. 2b. Indicate percentage of positive cells on each day. Include isotype control antibody for reference. Also compare with primary macrophages.

2. Extended Data Fig. 2i. Images for VE-cadherin staining of iVECs should be done with confluent cells to demonstrate localization to EC-EC junctions.

3. Extended Data Fig. 2j. Indicate percentage of HAECs and iVECs which took up acetylated LDL.

4. Figure 1d. While beta-galactosidase mean fluorescence intensity is higher for the WS-IM β , the effect size appears modest based on the flow cytometry results. Verify percentage of cells which are positive for beta-galactosidase.

5. Figure 2e. Since it appears that fewer WS-iVSMCs are positive for calponin even in the absence of oxLDL or macrophages, quantify calponin positive SMCs per unit area to validate claim that the WS-macrophages reduce calponin positive cells. Separate labeling of macrophages and conditioned media experiments could be used to determine whether direct contact of macrophages is needed to affect SMC calponin levels in the WS-iVSMCs.

6. Fig. 5a, the y-axis should be multiplied by 100 to represent percent efficiency.

7. Line 300. Provide a citation for the statement "Type I IFN signaling was recently recognized as an inducer of atherosclerosis."

Reviewer #2 (Remarks to the Author):

The manuscript entitled "Retrotransposons in Werner syndrome-derived macrophages trigger type I interferon-dependent inflammation in an atherosclerosis model" by Paul et al, describe functional analysis of iPSC derived macrophages and vascular cells from patients with Werner Syndrome and healthy controls. Their results demonstrate lower proliferation capacity and increased senescence of WS-derived macrophages, and characteristics of endothelial dysfunction and VSMC phenotypic changes in co-culture assays. In addition, the authors provide evidence of induced type I interferon signaling in WS-derived macrophages that could be attributed to the increased expression of retrotransposable elements. Although linking the Werner Syndrome to derepression of RTEs and activation of IFN-I is a

novel and interesting finding, the results in their current format are insufficient to support the claims or to provide a conclusive mechanism.

1. One major limitation of the study comes from the fact that all analysis is based on 2-3 different WS patients and healthy donors. Similar changes could be seen by randomly selecting 3 healthy donors (i.e. some donors showing more senescence; senescent phenotype has been generally linked with LINE derepression and activation of IFN-I (PMID: 30728521)). Isogenic controls with gene correction to restore WRN expression (e.g. PMID: 32320127) would allow for more reliable conclusions.

2. It is unclear how much of the phenotypic changes of VECs and VSMCs are contributed by the co-culture of the cells with macrophages and how much by the fact that VECs and VSMCs also originate from WS-iPSCs. To specifically demonstrate the contribution of WS-derived macrophages, the authors need to present data where the VECs and VSMCs also come from the healthy donors.

3. It is unclear how the RNA-Seq analysis was done in Figure 3. What were the groups that were compared here, one group against all or WS vs healthy? The clustering plot is rather confusing, and it would be more clear if the authors provided pairwise analysis of the WS vs healthy in both conditions. Also, the complete top gene ontology information for all the up- and downregulated lists (currently the 4 clusters) should be provided for unbiased (not selected by authors) analysis.

4. In Figure 4, the ATAC-Seq data is based on two replicates of which the healthy donors show large variation compared to WS-derived cells. Actually, the two healthy samples show as much variation as the healthy vs WS (in line with point #1) which makes it impossible to draw reliable conclusions with just n of 2. Why were the healthy_1 and WS_2 removed from the analysis? From the analysis provided, it is hard to evaluate what is causing the larger amount of DE peaks being specific to healthy macrophages, is it quality of the data, number of the peaks etc? The authors should report the number of peaks identified in each condition and present their overlap (e.g. venn). As the sequencing data is not provided to the Reviewers it is also impossible to evaluate the quality of the data. Therefore, the results presented in their current format are insufficient to support the conclusions drawn from the data.

5. Further providing a link between RTE derepression and WRN could be worth pursuing to strengthen this major finding of the manuscript. Does knockdown of WRN expression in healthy cells or restoration in WS cells, affect RTE expression and/or IFN-I activation? Or is this link solely exerted through the senescence phenotype as previously shown (PMID: 30728521)?

6. Please provide references to all the computational tools/packages used in the Methods (e.g. DESeq2, HOMER etc)

Reviewer #3 (Remarks to the Author):

Review

This is a most interesting paper that links retrotransposons that are up-regulated in Werner (WRN) Syndrome to a type 1 interferon response and to consequent phenotypic changes in iSPC derived macrophages M \emptyset (iM \emptyset) that include apoptosis, senescence and heightened cytokine secretion. The findings are based upon iM \emptyset , with some phenotypic data on iSMC and iEC from WRN versus controls particularly in response to activated control M \emptyset . An increase in some of the retrotransposons is also shown in the circulating monocytes from Werner syndrome patients. While there is considerable disease modeling that reflects a novel abnormal phenotype in the Werner vs. control cells and insight into a new mechanism, the reason for the increase in retrotransposons in response to the Werner mutation is not elucidated. Does type 1 interferon inhibition block the SMC and EC phenotype in co-culture? Since WRN is a DNA repair gene does unrepaired DNA damage lead to an increase in retrotransposons that cause a type I interferon response? Please see specific suggestions. Does oxLDL increase retrotransposons?

Abstract:

The data linking the type I interferon response with the phenotype of the iM \emptyset should be mentioned, ie. the cytokine, senescent or apoptotic data?

Was the interferon signature in the iM \emptyset or also in the iVEC and iVSMC? Were the retrotransposons increased in those cells as well or in iM \emptyset treated with oxLDL?

What kind of phenotypic switch was observed in the iVEC and iSMC cells \pm co-culture with iM \emptyset ?

What is the connection between WRN gene and the increase in retrotransposons?

Introduction:

The introduction should describe evidence of heightened expression of retrotransposons in myeloid cells, M \emptyset , monocytes and neutrophils associated with vascular pathology, see Saito et al, *Circulation* 135:1920-35, 2017 and more recently, Taylor et al in *Amer J Resp Crit Care Med*, June 13 online

Results:

Line 117: Extended Data Figure 2k. The SMC look cuboidal in the phase contrast but spindle shaped in the IHC studies. Was the phase contrast an earlier stage of maturation. Please comment.

Line 135-142: Are there two populations of cells, one that is senescent and one that is pro-apoptotic? Are these phenotypes related to the amount of DNA damage? P53 expression?

Figure 1: Why is oxLDL necessary to bring out the inflammatory phenotype but not the senescent phenotype since they should go together?

Figure 2: It is unclear whether the oxLDL activated iMØ that impacted the WRN iVEC were from WRN iMØ. The same is true for the SMC. So we should see data from iMØ control and WRN and iVEC control and WRN to determine whether there is an additive effect or synergism. The same is true for iSMC and iMØ.

Figure 3: More data need to be given to support Figure 3. All the clusters are interesting to show. If oxLDL iMØ behave like untreated WRN iMØ in f and g why is this not reflected in b?

What about senescent genes? Transcriptomic data on iEC and iSMC would also be valuable in interpreting the interactions.

Figure 4: Given the dramatic effect of oxLDL activation even in controls in cluster 4, it is strange that there were no differentially accessible regions. How is this explained? At least a subset should be common to activated control and WRN iMØ. Does oxLDL not increase retrotransposons and the consequent IFN cascade? This should be spelled out.

Figure 5: Are these properties exclusively those of iMØ or is there evidence that an increase in retrotransposons and double stranded RNA is also a feature of iVEC and iVSMC from WRN patients?

Figure 7: The Figure does not explain the increase in retrotransposons as a consequence of WRN. The authors should determine whether there is increased DNA damage in the WRN iMØ and whether repair of DNA damage can reverse the increase in retrotransposons and the subsequent induction of ORF3/7 mediated phenotype related to apoptosis, senescence and heightened cytokine production. This certainly needs to be discussed to link the mutation to the pathogenesis. What is it about unrepaired DNA that leads to de-methylation of retrotransposons that are normally silenced.

REVIEWER COMMENTS

Reviewer #1

This study presents some potentially novel observations about the mechanism by which macrophages in Werner syndrome may influence atherosclerosis. The authors use coculture of iPS-derived macrophages and either vascular smooth muscle cells (iVSMCs) or vascular endothelial cells (iVECs) to demonstrate potential differences in the function of healthy and Werner syndrome (WS) macrophages that might be relevant in atherosclerosis. The most intriguing result is that the WS macrophages may have an activated response to type I IFN-related genes due to retrotransposon expression. As noted in major comments, the effect sizes are modest in a number of cases and may depend on the presence of oxidized LDL (oxLDL), so additional studies are needed to validate the study and increase the overall significance.

Major Comments

1. While the first two paragraphs of the introduction are accurate, they should focus on Werner syndrome and its pathology since that is the focus of this study. Also, in the Introduction, provide evidence for the, address the role of type I interferons in atherosclerosis.

Response: We rewrote the Introduction section to place more emphasis on Werner syndrome and added references to studies investigating the role of type I IFN in atherosclerosis.

2. Assay for differences in monocyte receptor adhesion molecule VCAM-1 on Healthy- and WS-iVEC to address whether this explains difference in macrophage adhesion.

Response: We confirmed that VCAM-1 expression was not changed (data not shown) but ICAM-1 expression was upregulated in the presence of WS-iMφs regardless of oxLDL treatment (revised Fig. 2b,c). We also confirmed the greater tendency for WS-iMφs without oxLDL to attach to VECs compared with healthy-iMφ (Fig. 2a,c middle). Moreover, oxLDL-treated WS-iMφs appear to affect VECs and increase ICAM-1 expression (Fig. 2a-c). Therefore, we conclude that WS-iMφs attach to VECs through ICAM-1 expression.

3. Except for TNF-alpha mRNA and protein expression, the mRNA expression of cytokines in coculture with iVECs appears to depend on the presence of oxLDL. Performing the experiment with healthy iVECs and WS macrophages could identify a specific WS-macrophage effect. Further, since there are two types of macrophages and responses are measured with and without oxLDL, a two factor ANOVA should be performed. An interaction effect should arise from the two-factor ANOVA.

Response: We performed cross-co-culture experiments revealing that both WS-iMφ/WS-iVEC and WS-iMφ/healthy-iVEC combinations showed increased iMφ adhesion on iVECs, ICAM-1 induction, and IL6 and TNF-alpha release to the same extent (Supplementary Fig. 5a-d). These findings suggest

that WS-iM ϕ s have strong inflammatory effects on endothelial cells. By contrast, there was no clear difference between healthy-iM ϕ /healthy-iVECs and healthy-iM ϕ /WS-iVECs (Extended Fig. 5a-d). In addition, we performed RNA-seq analysis of healthy- and WS-iVECs and found no inflammatory pathway enrichment in WS-iVECs; therefore, we conclude that WS-iVECs had no major abnormalities (Supplementary Fig. 6a-e).

We did perform two-way ANOVA but did not mention this in the Figure legends in the original manuscript. We now make note of this statistical test in the revised Fig. 2 legend and Methods section.

4. Based on mRNA and contractile protein expression, reduced contractile protein demonstrated, but no evidence is provided that iVSMCs are synthetic. (lines 169-170).

Response: Specific markers that are upregulated in the synthetic phenotype are rare. Instead, the disappearance of proteins associated with the contractile phenotype is generally taken as characteristic of the synthetic phenotype (Rensen et al., *Neth Heart J.*, 2007). Aberrant cell proliferation is also a hallmark of the SMC synthetic phenotype (Beamish et al., *Tissue Eng Part B Rev.*, 2010). Therefore, we focused on contractile marker expression and aberrant cell proliferation to assess the effect of inflammatory macrophages on VSMCs. At least, we can claim that iVSMC, after iM ϕ stimulation, loses contractile characteristics. We added these references to the revised manuscript.

5. iVSMC proliferation and reduced gene expression of SMC contractile proteins depends on oxLDL with WS-macrophages. This appears to be due to an interaction effect of the WS-macrophages and oxLDL. At the least, these statements need to be qualified (lines 162-163 and Fig. 2c).

Response: Werner syndrome iPS-derived VSMCs showed aberrant proliferation and reduced contractile marker expression compared with healthy-iVSMCs when co-cultured with inflammatory M ϕ s. In our co-culture system, after three washes with PBS, we seeded untreated or oxLDL-treated M ϕ s onto iPS-derived VSMCs. In this experiment, no additional oxLDL was added to the coculture. Therefore, the interaction effects observed in this experiment arose only from the accelerated inflammatory state of macrophages treated with oxLDL, which induced phenotypic changes to iPS-derived VSMCs, and were not the result of a direct effect of oxLDL on VSMCs. We now emphasize that no additional oxLDL was added to the co-culture in the revised Methods section.

6. With respect to the upregulation of endogenous type I IFN signals in WS-macrophages relative to healthy-macrophages (Lines 190-195, Fig 3g), there is some suppression due to oxLDL and it is unclear that in the presence of oxLDL that there is a difference between the healthy and WS macrophages. Statistical comparison of these conditions is needed to validate the claim. In contrast, apoptosis genes are increased in WS-macrophages after addition of oxLDL relative to healthy case.

Response: We added samples derived from *WRN* gene-corrected iM ϕ s and reanalyzed all RNA-seq samples together. We also performed GSEA for pairwise comparisons and calculated statistical

significance. When healthy-iMφs were stressed with 50 μg/ml oxLDL alone, we confirmed some RTE de-repression (Fig. 6a, cluster 1 and 4 (C1 and C4)) and upregulation of IFN signaling, inflammation-related genes, and apoptosis-related genes at the transcription level (Fig. 3g; Extended Data Table 1).

On the other hand, *WRN* mutation induced RTE de-repression and RTE-dependent upregulation of IFN signaling, which led to aberrant cell cycle status, apoptosis, and senescence (Fig. 3c,e and Fig. 5; Extended Data Table 1). Even without oxLDL, WS-iMφs are already in the later phase of senescence (Fig. 3c) (Nakao et al., *Trend Cell Biol.* 2020). Because IFN signal genes are already upregulated in WS-iMφs, changes in RTE and IFN signal genes after oxLDL treatment in WS cells were relatively modest compared with those in healthy cells (Fig. 3h, Fig. 6a,d). However, after treating WS-iMφs with oxLDL, other inflammatory signals (i.e., apoptosis pathway, NF-kappa, TNF, and oxidative stress) were increased (Extended Data Table 1, Fig. 3h).

A recent study reports that inflammatory signals induced by LPS and lipids cooperatively accelerate inflammatory signals in peripheral blood-derived monocytes/macrophages (Oishi et al., *JCI Insight*, 2022). Like LPS treatment, RTE-induced inflammatory signals are already active in WS cells in the absence of oxLDL, and SASP was accelerated by oxLDL treatment. On the other hand, only oxLDL treatment in healthy-iMφ cells resulted in milder SASP.

The statistical significance of each pairwise comparison is now reported in the revised manuscript. P-values were derived from Cuffdiff analysis.

7. To show that the knockdown of IRF3/7 does not affect type I IFN gene signatures, senescence, and cytokines expression is specific to WS-macrophages, similar knockdown experiments need to be performed with healthy macrophages.

Response: We performed IRF3/7 knockdown in healthy- and gcWS-iMφs and confirmed no significant effects on these measures. We speculate that this is because healthy- and gcWS-iMφs without oxLDL treatment did not show upregulation of type I IFN genes. Therefore, we conclude that RTE-dependent type I IFN signal gene upregulation and consequent cell proliferation inhibition were specific to WS-iMφs. We added these data to Supplementary Fig.7 (healthy-iMφs: a-g, gcWS-iMφs: h-l).

8. Lines 210-215. Show level in WS-macrophages relative to healthy macrophages of enrichment of IRF and other sites noted ATAC-seq motif enrichment analysis. (fig. 4).

Response: We now report the log p-values of IRFs and inflammatory motifs in Fig. 4g.

9. Likewise to support the specificity of the knockdown of DHX58 on IRF7, ISG15, MX1, and MX2 in WS-macrophages, perform similar experiments with healthy macrophages (Lines 280-285 and Fig. 7).

Response: We performed *DHX58* knockdown in healthy-iMφs and confirmed no significant difference in *IRF7*, *ISG15*, *MX1*, or *MX2* mRNA expression, as healthy-iMφs did not show aberrant reactivation of RTEs (Fig. 6) or accumulation of dsRNA (Fig. 7b). These new data were added to Fig. 7d,f.

Specific Comments.

1. Supplementary Fig. 2b. Indicate percentage of positive cells on each day. Include isotype control antibody for reference. Also compare with primary macrophages.

Response: We added the percentage of positive cells each day and an unstained control. We also added data from primary macrophages (Supplementary Fig. 2b).

2. Supplementary Fig. 2i. Images for VE-cadherin staining of iVECs should be done with confluent cells to demonstrate localization to EC-EC junctions.

Response: We re-stained iVECs with anti-VE-cadherin and confirmed that VE-cadherin localized in the EC-EC junction. We replaced the old images with new images in Supplementary Fig. 2i.

3. Supplementary Fig. 2j. Indicate percentage of HAECs and iVECs which took up acetylated LDL.

Response: We added mean fluorescence intensity (MFI) to Supplementary Fig. 2j.

4. Figure 1d. While beta-galactosidase mean fluorescence intensity is higher for the WS-IM, the effect size appears modest based on the flow cytometry results. Verify percentage of cells which are positive for beta-galactosidase.

Response: When we added data from unstained control cells, almost all cells in healthy, WS, and gcWS groups were shifted to the right compared with the unstained control group. The beta-galactosidase staining kit that we used cannot clearly discriminate between positive and negative populations; rather, the entire population shifts due to endogenous β -galactosidase activity (Nishizawa et al., *Cell Death Dis.* 2021). Therefore, as it is difficult to perform comparisons based on the percent of positive cells, we used mean fluorescence intensity (MFI). In Fig. 1g, the fluorescence histogram in the WS group is shifted to the far right compared with the other two groups.

5. Figure 2e. Since it appears that fewer WS-iVSMCs are positive for calponin even in the absence of oxLDL or macrophages, quantify calponin positive SMCs per unit area to validate claim that the WS-macrophages reduce calponin positive cells. Separate labeling of macrophages and conditioned media experiments could be used to determine whether direct contact of macrophages is needed to affect SMC calponin levels in the WS-iVSMCs.

Response: In this particular experiment, conditioned media with oxLDL cannot be used because the media itself contains oxLDL. Therefore, the effect observed by using conditioned media with oxLDL would be the combined effect of oxLDL and inflammatory foam Mφs. Therefore, we examined the

effect of conditioned media in non-treated iMφs. To examine the effect of conditioned media from iMφs, we used non-treated iMφs and found no significant differences in contractile marker gene expression among healthy, WS, and gcWS groups (Supplementary Fig. 5k). In addition, oxLDL alone did not induce changes in contractile markers (Supplementary Fig. 6f-j). These findings suggest that the effect of oxLDL-induced inflammatory foam WS-iMφs is essential for reducing contractile markers in WS-iVSMCs.

We used mean fluorescence intensity (MFI) to quantify calponin-1 expression (Fig. 2h).

6. Fig. 5a, the y-axis should be multiplied by 100 to represent percent efficiency.

Response: We multiplied the y-axis by 100.

7. Line 300. Provide a citation for the statement “Type I IFN signaling was recently recognized as an inducer of atherosclerosis.”

Response: We added a citation to support this statement.

Reviewer #2

The manuscript entitled “Retrotransposons in Werner syndrome-derived macrophages trigger type I interferon-dependent inflammation in an atherosclerosis model” by Paul et al, describes the functional analysis of iPSC-derived macrophages and vascular cells from patients with Werner Syndrome and healthy controls. Their results demonstrate lower proliferation capacity and increased senescence of WS-derived macrophages and characteristics of endothelial dysfunction and VSMC phenotypic changes in co-culture assays. In addition, the authors provide evidence of induced type I interferon signaling in WS-derived macrophages that could be attributed to the increased expression of retrotransposable elements. Although linking the Werner Syndrome to derepression of RTEs and activation of IFN-I is a novel and interesting finding, the results in their current format are insufficient to support the claims or to provide a conclusive mechanism.

1. One major limitation of the study comes from the fact that all analysis is based on 2-3 different WS patients and healthy donors. Similar changes could be seen by randomly selecting 3 healthy donors (i.e. some donors showing more senescence; senescent phenotype has been generally linked with LINE derepression and activation of IFN-I (PMID: 30728521)). Isogenic controls with gene correction to restore WRN expression (e.g. PMID: 32320127) would allow for more reliable conclusions.

Response: We repaired *WRN* gene mutation using the CRISPR-Cas9 system and established three gene-corrected (gc)WS-iPSCs (Kato et al., *Stem Cell Res.* 2021). We corrected one allele of the *WRN* gene locus and confirmed that around 51.22% of *WRN* mRNA expression was recovered in gcWS-iPS-derived Mφs (Fig. 1a). Impaired proliferation, an accelerated senescence phenotype, increased

inflammation-related gene expression, and de-repression of some WS-specific RTEs were also partially recovered in gcWS-iMφs (Fig. 1, 3, 4, and 6).

2. It is unclear how much of the phenotypic changes of VECs and VSMCs are contributed by the co-culture of the cells with macrophages and how much by the fact that VECs and VSMCs also originate from WS-iPSCs. To specifically demonstrate the contribution of WS-derived macrophages, the authors need to present data where the VECs and VSMCs also come from the healthy donors.

Response: To elucidate the specific impact of WS-iMφs, we performed a cross-co-culture experiment using iVECs and iVSMCs sourced from healthy donors. In this experiment, both WS-iMφ/WS-iVEC and WS-iMφ/healthy-iVEC combinations showed increased iMφ adhesion on iVECs, ICAM-1 induction, and IL6 and TNF-alpha release to the same extent in iVECs, suggesting that WS-iMφs have strong inflammatory effects on both healthy- and WS-iVECs (Supplementary Fig. 5a-d). In addition, we performed RNA-seq analysis of healthy- and WS-iVECs and found no inflammatory pathway enrichment in WS-iVECs, indicating that WS-iVECs had no major abnormalities (Supplementary Fig. 6a-e). However, only the WS-iMφ/WS-iVSMC combination induced phenotypic changes in iVSMCs (Supplementary Fig. 5e-f). We also performed RNA-seq analysis of healthy- and WS-iVSMCs without iMφs and found no inflammatory pathway enrichment in WS-iVSMCs (Supplementary Fig. 6f-j). Therefore, we conclude that WS-iVSMCs showed abnormalities only in the presence of WS-iMφs.

3. It is unclear how the RNA-Seq analysis was done in Figure 3. What were the groups that were compared here, one group against all or WS vs healthy? The clustering plot is rather confusing, and it would be more clear if the authors provided pairwise analysis of the WS vs healthy in both conditions. Also, the complete top gene ontology information for all the up- and downregulated lists (currently the 4 clusters) should be provided for unbiased (not selected by authors) analysis.

Response: We conducted gene set enrichment analysis (GSEA) to perform pairwise comparisons and plotted normalized enrichment scores (NES) of the top 10 significantly enriched pathways in each group at maximum. The corresponding genes of interest were picked among the top 10 significantly enriched pathways and used to compose heatmaps with log₂ of FPKM values. P-values were derived from pairwise comparisons in Cuffdiff.

4. In Figure 4, the ATAC-Seq data is based on two replicates of which the healthy donors show large variation compared to WS-derived cells. Actually, the two healthy samples show as much variation as the healthy vs WS (in line with point #1) which makes it impossible to draw reliable conclusions with just n of 2. Why were the healthy_1 and WS_2 removed from the analysis? From the analysis provided, it is hard to evaluate what is causing the larger amount of DE peaks being specific to healthy macrophages, is it quality of the data, number of the peaks etc? The authors should report the number of peaks identified in each condition and present their overlap (e.g. venn). As the sequencing data is not

provided to the Reviewers it is also impossible to evaluate the quality of the data. Therefore, the results presented in their current format are insufficient to support the conclusions drawn from the data.

Response: We reanalyzed three healthy-iMφ, four WS-iMφ, and three gcWS-iMφ samples with or without oxLDL treatment. As we did not observe any consistent differences between oxLDL-treated and non-treated iMφs, we analyzed the samples regardless of oxLDL treatment (healthy-iMφs: 6 samples, WS-iMφs: 8 samples, gcWS-iMφs: 6 samples).

Again, we confirmed that the number of differentially accessible regions (DARs) was larger in healthy-iMφs than in WS-iMφs (Fig. 4a). Healthy-iMφ-specific DARs were enriched in cell cycle and cell differentiation-related factor binding motifs (E2F7, MYB, PBX1, and GATA; Fig. 4e). On the other hand, WS-iMφ-specific DARs were enriched in cell interferon and inflammation-related factor binding motifs (IRF, JUN-API, and CEBP; Fig. 4e). These WS-specific chromatin signatures were partly recovered by *WRN* gene correction (number of DARs, Fig. 4b; inflammation and cell cycle-related factor binding motifs, Fig. 4f).

5. Further providing a link between RTE derepression and *WRN* could be worth pursuing to strengthen this major finding of the manuscript. Does knockdown of *WRN* expression in healthy cells or restoration in WS cells, affect RTE expression and/or IFN-I activation? Or is this link solely exerted through the senescence phenotype as previously shown (PMID: 30728521)?

Response: Based on the reviewer's earlier recommendation, we restored *WRN* in WS cells instead of knocking down *WRN* expression in healthy cells. We analyzed the expression of RTEs in healthy-, WS- and gcWS-iMφs with or without oxLDL treatment. To systematically examine global RTE expression in iMφs, we used RNA-seq analysis. We identified the genomic regions representing RTEs and used the RNA read counts within these regions in each sample to identify differentially expressed RTEs. We performed pairwise analysis of WS- or gcWS-iMφs with healthy-iMφs to detect upregulated RTEs in WS- and gcWS-iMφs. We found that among 101 upregulated RTEs in gcWS-iMφs compared with healthy-iMφs, 100 were common with WS-iMφs (Fig. 6b). Interestingly, we confirmed that RTE expression was reduced in gcWS-iMφs compared with WS-iMφs (Fig. 6a, cluster 5-7; Fig. 6d).

Furthermore, to unveil why WS-iMφs cells showed significantly higher RTE expression, we performed H3K9me3 ChIP-seq. RNA-seq pairwise analysis revealed that individual numbers of RTEs, even at the sub-family level, were higher in WS-iMφs compared with healthy-iMφs. Consistently, H3K9me3 levels in these regions were significantly lower in WS-iMφs than in healthy-iMφs (Fig. 6e). As a previous study reports that Suv39h-dependent H3K9me3 chromatin specifically represses intact LINE elements in the mouse embryonic stem cell epigenome (Bulut-Karslioglu et al., *Mol Cell*. 2014), this suggests that loss of H3K9me3 levels is correlated with RTE de-repression in WS-iMφs.

However, we observed no significant difference in H3K9me3 levels between WS- and gcWS-iMφs (Fig. 6f). Thus, we could not reach a clear conclusion as to why RTE expression was reduced in gcWS-iMφs compared with WS-iMφs. We believe there may be a negative feedback loop at play here: reduced

H3K9me3 levels de-repress RTEs, but if RTEs become active and start moving around the genome, this can potentially disrupt the epigenetic landscape, including H3K9me3 levels, in the affected regions. In our study, gcWS-iMφs showed reduced RTE and IFN signal gene expression but did not show changes at the H3K9me3 level. These data may indicate that de-repressed RTEs disrupt the epigenetic landscape and reduce the expression of Suv39h and H9A, thereby affecting H3K9me3 levels, which could not be recovered in gcWS-iMφs because they still displayed higher RTE expression than healthy-iMφs. Alternatively, the decrease in H3K9me3 levels may not be the primary mechanism of WRN mutation but rather an outcome. In fact, we also confirmed that the expression of several heterochromatin signature genes did not differ between WS- and gcWS-iMφs (shown below). It is possible that epigenetic changes may be a secondary event in this case, which will be a focus of future studies.

Figure: mRNA expression pattern of several heterochromatin signature genes.

6. Please provide references to all the computational tools/packages used in the Methods (e.g. DESeq2, HOMER etc)

Response: We now mention all computational tools/packages used in the revised manuscript.

Reviewer #3

This is a most interesting paper that links retrotransposons that are up-regulated in Werner (WRN) Syndrome to a type 1 interferon response and to consequent phenotypic changes in iSPC derived macrophages MØ (iMØ) that include apoptosis, senescence and heightened cytokine secretion. The findings are based upon iMØ, with some phenotypic data on iSMC and iEC from WRN versus controls particularly in response to activated control MØ. An increase in some of the retrotransposons is also shown in the circulating monocytes from Werner syndrome patients. While there is considerable disease modeling that reflects a novel abnormal phenotype in the Werner vs. control cells and insight into a new mechanism, the reason for the increase in retrotransposons in response to the Werner mutation is not elucidated

>Does type 1 interferon inhibition block the SMC and EC phenotype in co-culture?

Response: We inhibited type I IFN by knocking down IRF3/7 in WS-iMφs, treating them with oxLDL, and co-culturing them with WS-iVECs or WS-iVSMCs and found that the phenotypes were partially rescued (Supplementary Fig. 8). Consistently, Mφ-iVEC adhesion as well as cell surface expression of ICAM-1 and *ICAM-1* mRNA were also reduced (Supplementary Fig. 8a-c). After IRF3/7 knockdown, WS-iVSMCs co-cultured with WS-iMφs displayed a trend toward increased contractile gene expression (Supplementary Fig. 8d).

>Since WRN is a DNA repair gene does unrepaired DNA damage lead to an increase in retrotransposons that cause a type I interferon response? Please see specific suggestions.

Response: We confirmed that WS-iMφs exhibited a DNA damage signal (Fig. 1b,c) and RTE reactivation and that these aberrant phenotypes were reset in gcWS-iMφs (Fig. 6a-d), indicating that *WRN* mutation induces a DNA damage signal and RTE reactivation in macrophages. However, we could not draw conclusions about the relationship between the DNA damage signal and RTE reactivation in this study.

Aberrantly accumulated self or foreign DNA in cytoplasm is readily detected by cGAS. Active cGAS catalyzes ATP and GTP into 2'3'-cGAMP, which then binds to and activates the adaptor protein STING, followed by initiated transcription of type I IFN signature genes via TBK1 and IRF3 (Ma et al. *FASEB J.* 2020). However, we observed no notable change in cGAS (*MB21D1*) or STING (*TMEM173*) mRNA expression in WS-iMφs compared with healthy-iMφs (Supplementary Fig. 11b). Moreover, we found significantly higher accumulation of double-stranded (ds) RNA (Fig. 7b) and mRNA expression of its sensor, *DHX58*, in WS-iMφs compared with healthy-iMφs (Fig. 3c). We also confirmed that *DHX58* inhibition significantly reduced the expression of type I IFN signature genes only in WS-iMφs but not in healthy-iMφs (Fig. 7e-f).

For further confirmation, we blocked the reverse transcription of RNA to DNA using two nucleoside/nucleotide reverse transcriptase inhibitors, lamivudine and emtricitabine, but observed no notable changes in the expression of cGAS, STING, or type I IFN signature genes (Supplementary Fig. 10c-e).

Therefore, we conclude that RTE-derived dsRNA, rather than WRN-induced damaged DNA, evoked the type I IFN response via the *DHX58* nucleic acid sensor.

>Does oxLDL increase retrotransposons?

Response: In healthy-iMφs, some RTEs were de-repressed after oxLDL treatment. We identified three clusters (clusters 1, 2, and 4) that were dynamically changed after oxLDL treatment (Fig. 6a). RTEs in cluster 2 were suppressed and those in clusters 1 and 4 were upregulated after oxLDL treatment only in healthy-iMφs. WS-iMφs showed less of a response to oxLDL treatment than healthy-iMφs in terms of both RTE activation and IFN signal gene expression (Fig. 3, Fig. 6a).

Abstract:

>The data linking the type I interferon response with the phenotype of the iMØ should be mentioned, ie. the cytokine, senescent or apoptotic data?

Response: We added the finding that IRF3/7 knockdown in WS-iMφs resulted in proliferation recovery, reduced senescence, and inflammatory cytokine production (as shown in Fig. 5) to the Abstract.

>Was the interferon signature in the iMØ or also in the iVEC and iVSMC? Were the retrotransposons increased in those cells as well or in iMØ treated with oxLDL?

Response: To answer this question, we performed RNA-seq analysis in healthy- and WS-iVECs and -iVSMCs. We did not observe upregulation of IFN signal genes in WS-iVECs or WS-iVSMCs (Extended Fig. 6a-j). We also analyzed the pattern of RTE expression in iVECs and iVSMCs; however, there were no differentially expressed RTEs between healthy and WS groups, regardless of oxLDL treatment (mentioned in the main text but not shown in a figure). The question about iMφs is answered above.

>What kind of phenotypic switch was observed in the iVEC and iSMC cells± co-culture with iMØ?

Response: In co-culture, WS-iMφs induced endothelial dysfunction in WS-iVECs and characteristics of the synthetic phenotype in WS-iVSMCs. This is shown in Fig. 2 and explained in the Results section.

>What is the connection between WRN gene and the increase in retrotransposons?

Response: We restored *WRN* in WS cells and analyzed the expression of RTEs in healthy-, WS-, and gene-corrected (gc)WS-iMφs with or without oxLDL treatment. To systematically examine global RTE expression in iMφs, we used RNA-seq analysis. We identified the genomic regions representing RTEs and used RNA read counts within these regions in each sample to identify differentially expressed RTEs.

Next, we performed pairwise comparisons of WS- or gcWS-iMφs with healthy-iMφs to identify upregulated RTEs in WS- and gcWS-iMφs. The numbers of genomic regions did not dynamically change between WS- and gcWS-iMφs. Among 101 upregulated RTEs in gcWS-iMφs, 100 were common with WS-iMφs (Fig. 6b). Interestingly, however, *WRN* gene correction significantly reduced RTE expression in gcWS-iMφs (Fig. 6a,d). Furthermore, to unveil why WS-iMφs cells showed significantly higher levels of RTE expression, we performed H3K9me3 ChIP-seq. RNA-seq pairwise analysis revealed that individual numbers of RTEs, even at the sub-family level, were higher in WS-iMφs than in healthy-iMφs. Consistently, H3K9me3 levels in those regions were significantly lower in WS-iMφs compared with healthy-iMφs (Fig. 6e) As a previous study reports that Suv39h-dependent H3K9me3 chromatin specifically represses intact LINE elements in the mouse embryonic stem cell epigenome (Bulut-Karslioglu et al. *Mol Cell*, 2014), this suggests that loss of H3K9me3 levels de-represses RTEs in WS-iMφs.

However, we observed no significant difference in H3K9me3 levels between WS- and gcWS-iMφs (Fig. 6f). Thus, we could not reach a clear conclusion as to why RTE expression was reduced in gcWS-iMφs compared with WS-iMφs. We believe there may be a negative feedback loop at play here: reduced H3K9me3 levels de-repress RTEs, but if RTEs become active and start moving around the genome, this can potentially disrupt the epigenetic landscape, including H3K9me3 levels, in the affected regions. In our study, gcWS-iMφs showed reduced RTE and IFN signal gene expression but did not show any change at the H3K9me3 level. These data may indicate that de-repressed RTEs disrupt the epigenetic landscape and reduce the expression of Suv39h and H9A, thereby affecting H3K9me3 levels, which could not be recovered in gcWS-iMφs because they still displayed higher RTE expression than healthy-iMφs. Alternatively, the decrease in H3K9me3 levels may not be the primary mechanism of *WRN* mutation but rather an outcome. In fact, we also confirmed that the expression of several heterochromatin signature genes did not differ between WS- and gcWS-iMφs (shown below). It is possible that epigenetic changes may be a secondary event in this case, which will be a focus of future studies.

Figure: mRNA expression pattern of several heterochromatin signature genes.

Introduction:

>The introduction should describe evidence of heightened expression of retrotransposons in myeloid cells, MØ, monocytes and neutrophils associated with vascular pathology, see Saito et al, *Circulation* 135:1920-35, 2017 and more recently, Taylor et al in *Amer J Resp Crit Care Med*, June 13 online

Response: We added this reference to the revised manuscript.

Results:

>Line 117: Extended Data Figure 2k. The SMC look cuboidal in the phase contrast but spindle shaped in the IHC studies. Was the phase contrast an earlier stage of maturation. Please comment.

Response: Thank you for noting this issue. We replaced the phase contrast image of iVSMCs in the revised manuscript.

>Line 135-142: Are there two populations of cells, one that is senescent and one that is pro-apoptotic? Are these phenotypes related to the amount of DNA damage? P53 expression?

Response: We confirmed that DNA damage was accumulated (Fig. 1b,c) and mRNA expression of *CDKN1A*, the target of p53 for apoptosis and senescence, was increased in WS-iMφs (Fig.1f). DNA damage induces both senescence and apoptosis through the p53-CDKN1A pathway (reviewed by Mijit et al., *Biomolecules*, 2020). We confirmed that both Annexin V⁺ cells and SA-beta-gal⁺ cells increased in WS-iMφs (Fig. 1e,g). Apoptotic cells and senescent cells do not always overlap; therefore, we believe there are at least two different populations in WS-iMφs.

>Figure 1: Why is oxLDL necessary to bring out the inflammatory phenotype but not the senescent phenotype since they should go together?

Response: Based on our data, we believe that the combined effect of oxLDL and heightened type I IFN signature genes induced by *WRN* mutation was important for releasing SASP.

oxLDL plays an important role in atherosclerosis development by inducing ROS production, inflammatory responses, and accumulation of lipids, which lead to fatty streak formation in vascular walls. On the other hand, type I IFN is a well-recognized inducer of cellular senescence (reviewed by Frisch and MacFawn, *Aging Cell*, 2020). Therefore, we believe that the combined effect of oxLDL and heightened type I IFN signature genes in WS-iMφs evoked an increase in SASP release (Fig. 1i). Although the difference was not significant compared with healthy- and gcWS-iMφ groups, non-treated WS-iMφs also showed increased pro-inflammatory secretion (Fig. 1i).

>Figure 2: It is unclear whether the oxLDL activated iMØ that impacted the WRN iVEC were from WRN iMØ. The same is true for the SMC. So we should see data from iMØ control and WRN and iVEC control and WRN to determine whether there is an additive effect or synergism. The same is true for iSMC and iMØ.

Response: Thank you very much for this recommendation. To elucidate the specific impact of WS-iMφs, we performed a cross-co-culture experiment using iVECs and iVSCMs sourced from healthy donors. In this experiment, both WS-iMφ/WS-iVEC and WS-iMφ/healthy-iVEC combinations showed increased iMφ adhesion on iVECs, ICAM-1 induction, and IL6 and TNF-alpha release to the same extent in iVECs, suggesting that WS-iMφs have strong inflammatory effects on both healthy- and WS-iVECs (Supplementary Fig. 5a-d). In addition, we performed RNA-seq analysis of healthy- and WS-iVECs and found no inflammatory pathway enrichment in WS-iVECs, indicating that WS-iVECs had no major abnormalities (Supplementary Fig. 6a-e). However, only the WS-iMφ/WS-iVSMC combination was required to induce phenotypic changes in iVSMCs (Supplementary Fig. 5e-f). We also performed RNA-seq analysis of healthy- and WS-iVSMCs without iMφs and found no inflammatory

pathway enrichment in WS-iVSMCs (Supplementary Fig. 6f-j). Therefore, we conclude that WS-iVSMCs showed abnormalities only in the presence of WS-iMφs.

>Figure 3: More data need to be given to support Figure 3. All the clusters are interesting to show. If oxLDL iMØ behave like untreated WRN iMØ in f and g why is this not reflected in b?

What about senescent genes?

Response: We first performed all pairwise comparisons among healthy-oxLDL(-), healthy-oxLDL(+), WS-oxLDL(-), and WS-oxLDL(+) in Cuffdiff and selected differentially expressed genes (DEGs) in at least one comparison. We then analyzed those differentially expressed genes with K-means clustering (shown in original Fig. 3). In the revised experiment, we added one WS-iMφ and three gene-corrected (gc)WS-iMφ samples and analyzed them in a similar way. Next, we performed gene set enrichment analysis (GSEA) for pairwise comparisons and plotted the normalized enrichment scores (NES) of the top 10 enriched pathways in each group. Finally, we picked the corresponding genes of interest among the top 10 enriched pathways and composed heatmaps with log2 of FPKM values. P-values were derived from the pairwise comparisons in Cuffdiff.

>Transcriptomic data on iEC and iSMC would also be valuable in interpreting the interactions.

Response: We added these data to Supplementary Fig. 6.

>Figure 4: Given the dramatic effect of oxLDL activation even in controls in cluster 4, it is strange that there were no differentially accessible regions. How is this explained? At least a subset should be common to activated control and WRN iMØ. Does oxLDL not increase retrotransposons and the consequent IFN cascade? This should be spelled out.

Response: We reanalyzed RNA-seq, RTE expression, and ATAC-seq with the addition of one WS-iMφ and three gene-corrected (gc)WS-iMφ samples. We confirmed that oxLDL treatment upregulated inflammation-related gene expression and RTE expression, especially in healthy-iMφs (Fig. 3 and 6). However, unexpectedly, there were no significant changes in chromatin-accessible regions after oxLDL treatment, even in healthy-iMφs (Fig. 4a). One possible explanation is that the chromatin regions related to inflammation, which are affected by oxLDL, were already in an open state prior to oxLDL treatment. Consequently, oxLDL may induce IFN signal gene expression irrespective of chromatin accessibility.

Figure 5: Are these properties exclusively those of iMØ or is there evidence that an increase in retrotransposons and double stranded RNA is also a feature of iVEC and iVSMC from WRN patients?

Response: We performed RNA-seq analysis in iVECs and iVSMCs derived from healthy- and WS-iPSCs and observed no inflammatory pathway enrichment in WS-iVECs or WS-iVSMCs (Extended Fig. 6); therefore, we conclude that WS-iVECs and WS-iVSMCs had no major abnormalities. In addition, we checked RTE expression patterns in these cells with RNA-seq. However, no differential

expression of RTEs was displayed by iVECs or iVSMCs (described in the main text but not shown in a figure). Moreover, we assessed the accumulation of dsRNA in iVECs and iVSMCs. However, no significant differences were observed (Supplementary Fig. 12).

Figure 7: The Figure does not explain the increase in retrotransposons as a consequence of WRN. The authors should determine whether there is increased DNA damage in the WRN iMØ and whether repair of DNA damage can reverse the increase in retrotransposons and the subsequent induction of IRF3/7 mediated phenotype related to apoptosis, senescence and heightened cytokine production. This certainly needs to be discussed to link the mutation to the pathogenesis. What is it about unrepaired DNA that leads to de-methylation of retrotransposons that are normally silenced.

Response: In the revised manuscript, we used recently established gene-corrected (gc)WS-iPSCs by repairing the genetic mutation in one of the alleles of the *WRN* gene locus using the CRISPR-Cas9 genome editing technique (Kato et al., *Stem Cell Res*, 2021). Gene correction partially rescued *WRN* mRNA expression and cell proliferation in gcWS-iMØs (Fig. 1a,d). In addition, apoptosis, senescence, inflammatory cytokine release, and the amount of damaged DNA were significantly reduced in gcWS-iMØs compared with WS-iMØs (Fig. 1).

In line with these results, transcriptomic (RNA-seq) and open chromatin (ATAC-seq) analysis revealed that type I IFN signature genes were enriched and inflammatory motifs like CEBP, JUN-AP1, IRFs, and ATF were found open in the WS-iMØ group, respectively (Fig. 3 and 4). Moreover, silencing type I IFN signaling in WS-iMØs rescued cell proliferation and suppressed cellular senescence and inflammation cytokine release (Fig. 7). Taken together, these results indicate that loss of *WRN* in WS-iMØs resulted in increased apoptosis, senescence, inflammatory cytokine release, and DNA damage, but some of these features were rescued in gcWS-iMØs. This means that *WRN* was responsible for all of these changes, including DNA damage.

Considering retrotransposable element (RTE) expression, gcWS-iMØs exhibited a subtle reduction in de-repressed RTEs compared with WS-iMØs (Fig. 6a, clusters 5, 6, and 7; Fig. 6d). A previous study reports that *WRN*-deficient mesenchymal stem cells show reduced H3K9me3 levels (Zhang et al., *Science*, 2015). In line with this study, we also found reduced H3K9me3 levels in WS-iMØs compared with healthy-iMØs. As Suv39h-dependent H3K9me3 chromatin specifically represses intact LINE elements (an RTE family member) in the mouse embryonic stem cell epigenome (Bulut-Karslioglu et al. *Mol Cell*, 2014), this suggests that loss of H3K9me3 levels derepresses RTEs in WS- and gcWS-iMØs.

Surprisingly, however, despite gcWS-iMØs displaying a 51.22% recovery of *WRN* mRNA expression and a reduced inflammatory phenotype, there was no notable difference in H3K9me3 level compared with WS-iMØs. Recently, Della Valle and colleagues conducted a study on LINE-1 in premature aging syndromes (i.e., Hutchinson-Gilford progeria syndrome and Werner syndrome). They observed that nuclear LINE-1 RNA expression occurred early in progeroid cells, which led to

suppression of histone-lysine N-methyltransferase SUV39H1 activity. This suppression resulted in the loss of heterochromatin and manifestation of senescent phenotypes. Depletion of LINE-1 RNA using antisense oligonucleotides restored epigenetic marks (i.e., H3K9me3 and H3K27me3) associated with heterochromatin and reduced the expression of genes associated with senescence in human cells and a mouse model of Hutchinson-Gilford progeria syndrome, ultimately extending the lifespan of mice (Della Valle et al. *Sci Transl Med.*, 2022). By contrast, Suv39h-dependent H3K9me3 chromatin specifically represses intact LINE elements in the mouse embryonic stem cell epigenome (Bulut-Karslioglu et al. *Mol Cell*, 2014). Thus, there may be a negative feedback loop at play here: reduced H3K9me3 levels derepress RTEs, but if RTEs become active and start moving around the genome, this can potentially disrupt the epigenetic landscape, including levels of H3K9me3 in the affected regions.

In our study, gcWS-iMφs showed reduced RTE and IFN signal gene expression but did not show any change in H3K9me3 level compared to WS-iMφs. These data may indicate that de-repressed RTEs disrupted the epigenetic landscape or decreased SUV39h expression, thereby affecting H3K9me3 levels, which could not be recovered in gcWS-iMφs because they still displayed higher RTE expression than healthy-iMφs. Alternatively, the decrease in H3K9me3 levels may not be the primary mechanism of WRN mutation but rather an outcome. These mechanisms will be clarified in future work.

Figure: mRNA expression pattern of several heterochromatin signature genes.

REVIEWERS' COMMENTS

Reviewer #1 (Remarks to the Author):

The authors have addressed all of my concerns and the manuscript is significantly improved.

Reviewer #2 (Remarks to the Author):

I appreciate the authors' thorough response to my previous comments and their effort in providing extensive additional data to support their findings. Thank you for this comprehensive revision. I have no further comments.

Reviewer #1 (Remarks to the Author):

The authors have addressed all of my concerns and the manuscript is significantly improved.

Reviewer #2 (Remarks to the Author):

I appreciate the authors' thorough response to my previous comments and their effort in providing extensive additional data to support their findings. Thank you for this comprehensive revision. I have no further comments.

Response:

We thank the reviewers for their comments and valuable feedback, which has improved the manuscript. We also would like to express our gratitude to Reviewer #1 and Reviewer #2 for considering the point-by-point response of Reviewer #3.